# Overcoming the Stability Gap in Continual Learning

**Md Yousuf Harun**                                                          *mh1023@rit.edu*
*Rochester Institute of Technology*

**Christopher Kanan**                                              *ckanan@cs.rochester.edu*
*University of Rochester*

**Reviewed on OpenReview:** *https://openreview.net/forum?id=o2wEfwUOma*

## Abstract

Pre-trained deep neural networks (DNNs) are being widely deployed by industry for making business decisions and to serve users; however, a major problem is model decay, where the DNN's predictions become more erroneous over time, resulting in revenue loss or unhappy users. To mitigate model decay, DNNs are retrained from scratch using old and new data. This is computationally expensive, so retraining happens only once performance significantly decreases. Here, we study how continual learning (CL) could potentially overcome model decay in large pre-trained DNNs and greatly reduce computational costs for keeping DNNs up-to-date. We identify the "stability gap" as a major obstacle in our setting. The stability gap refers to a phenomenon where learning new data causes large drops in performance for past tasks before CL mitigation methods eventually compensate for this drop. We test two hypotheses to investigate the factors influencing the stability gap and identify a method that vastly reduces this gap. In large-scale experiments for both easy and hard CL distributions (e.g., class incremental learning), we demonstrate that our method reduces the stability gap and greatly increases computational efficiency. Our work aligns CL with the goals of the production setting, where CL is needed for many applications [1].

## 1 Introduction

Deep neural networks (DNNs) are now widely deployed in the industry; however, a major problem for many companies is model decay, where the predictive performance of a DNN degrades over time (Zhou et al., 2020). This is primarily caused by concept drift (Tsymbal, 2004; Gama et al., 2014; Lu et al., 2018), where the nature of the predicted target variables changes over time, e.g., for a classifier, this would correspond to the introduction of new categories or an expanded definition individual classes. When model decay is detected, most companies employ *offline retraining (i.e., batch or joint retraining)*, where a model is retrained from scratch with a combination of old and new data (Egg, 2021). This is very expensive, so model monitoring is used to determine when offline training is required (Mäkinen et al., 2021). Despite frequent offline training, deployed models still suffer from accuracy drops of up to 40% (Mallick et al., 2022).

Continual learning (CL) is a promising solution for preventing model decay (Huyen, 2022a;b; Jain & Shenoy, 2022), where in CL the goal is to update a DNN incrementally with new data while preserving the previous knowledge (Parisi et al., 2019). In a GrubHub study, this enabled them to avoid model decay and provided a 45× decrease in training costs compared to daily offline training (Egg, 2021). To do this, GrubHub employed online updates on new samples. Online updates can cause catastrophic forgetting of past knowledge due to concept drift (McCloskey & Cohen, 1989), but due to the rapidly changing preferences of their customers, forgetting past knowledge was desirable. However, catastrophic forgetting is unacceptable for many industry applications because past knowledge must be retained. For CL to be widely adopted by industry, it must be shown to inhibit model decay in *pre-trained models*, provide significant computational benefits, work for

---

[1]Code is available at https://yousuf907.github.io/sgmsite

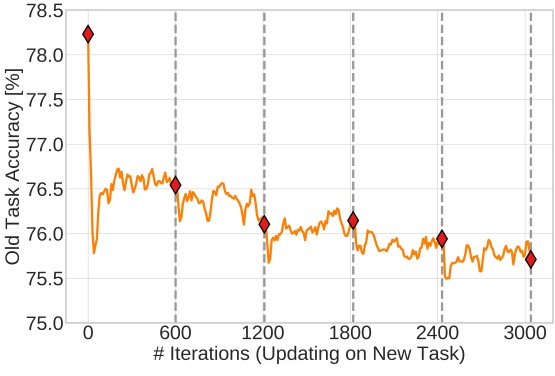

(a) Stability gap over all rehearsal sessions

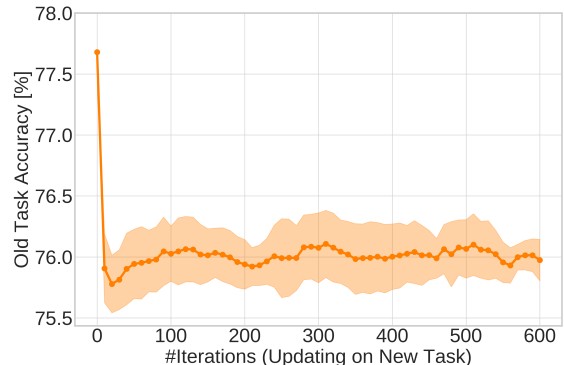

(b) Stability gap as an average over all rehearsals

Figure 1: **An overview of *stability gap* phenomenon.** The stability gap is a phenomenon that occurs in CL when learning new data, where accuracy on previously learned data (Y-axis) drops significantly as a function of training iterations when a new distribution is introduced (X-axis). Fig.(a) illustrates this behavior during CIL, where a network pre-trained on ImageNet-1K, learns 365 new classes from Places365-LT over five rehearsal sessions. Each rehearsal session involves 600 iterations that combine samples from the old and new tasks. A gray dotted vertical line marks the end of a rehearsal session or a task transition. When rehearsal begins, accuracy on the old task for the conventional rehearsal drops dramatically before slowly recovering, although it fails to recover the original performance on the old data. The traditional measures of catastrophic forgetting focus on performance at task transitions (red diamonds), ignoring significant forgetting that occurs during the learning process between task transitions. Fig.(b) shows the stability gap in the learning curve averaged over five rehearsal sessions. In this work, we attempt to mitigate the stability gap.

*arbitrary* shifts in concepts, and ideally be as effective as offline training. The majority of CL algorithms deviate from these criteria, i.e., they impose constraints on storage that are irrelevant to industry problems, shun pre-trained models, perform worse than offline retraining, design algorithms for only a single concept drift distribution (e.g., class incremental learning), and/or are more computationally expensive than offline retraining (Harun et al., 2023a;b; Prabhu et al., 2023b;a; Verwimp et al., 2024; Lee et al., 2023).

Here, we study using CL to mitigate model decay and to introduce new concepts into a DNN. Initially, we studied updating ImageNet-1K pre-trained models with new classes in class incremental learning (CIL) experiments. We used computationally constrained cumulative rehearsal to mitigate forgetting, where rehearsal involves mixing old samples with new ones to prevent forgetting, and cumulative rehearsal does this using all previously observed samples. Computational constraints were enforced by limiting the total number of rehearsal updates permitted. However, we found that CL's *stability gap* was a major obstacle to our goal (De Lange et al., 2023), as shown in Fig. 1.

The stability gap refers to the observation that when a model is updated on new data, accuracy on the classes observed in earlier batches plummets before rehearsal or other CL methods gradually recover old performance. In the typical CIL setting we adopted, samples arrive in batches where each batch contains classes only within that batch. As seen in Fig. 1, the performance plummets when learning a new batch during the rehearsal sessions and does not fully recover. The stability gap can be seen as *transient forgetting* due to introducing a new task. Typically, catastrophic forgetting refers to the performance drops observed at the end of learning new tasks (at task transitions). In contrast, the stability gap refers to the performance drops observed over learning steps (between task transitions). In our work, we seek to understand why the stability gap happens in our scenario and how to effectively mitigate it, enabling using fewer rehearsal updates while achieving higher accuracy.

**We test two hypotheses to examine the stability gap in CIL with pre-trained DNNs:**

1. ***The stability gap is increased in part due to having a large loss at the output layer for the new classes.*** To test this hypothesis, we study two methods to mitigate the large loss [2] in the output layer for the new classes. The first method is to initialize the output layer in a data-driven approach rather than randomly initializing the output units responsible for the new classes. The second method is a specialized form of soft targets for the network, rather than the typical hard targets used for the network, where these soft targets are designed to improve performance for the new classes while minimally perturbing others.

2. ***The stability gap is increased in part due to excessive network plasticity.*** We test this hypothesis by controlling the level of plasticity in network layers in a dynamic manner. For hidden layers, we test this hypothesis using LoRA (Hu et al., 2021), which reduces the number of trainable parameters in the hidden layers of the network. For rehearsal methods, after each rehearsal session, these weights are folded into the original network weights. For the output layer, we test this hypothesis by freezing the output units for classes seen in earlier batches during rehearsal.

In our experiments, we find that both hypotheses are supported. This leads us to develop a combined method that greatly reduces the stability gap for both CIL and other distributions.

**This paper makes the following major contributions:**

1. We are the first to study overcoming the stability gap. We test the aforementioned hypotheses in CIL and discover that they both play an important role. We propose novel metrics to measure the stability, plasticity, and continual knowledge gaps.

2. We are the first to study *maintaining or improving* performance on ImageNet-1K while learning additional classes, an aspect not investigated in previous research employing ImageNet-1K pre-trained backbones (Wang et al., 2022b;a; Smith et al., 2023; Gao et al., 2023; McDonnell et al., 2024). We study this for both the CIL and IID (independent and identically distributed) distributions. These experiments are conducted by combining ImageNet-1K with Places365 or Places365-LT (a combined 1365 total classes).

3. We develop a method that greatly mitigates the stability gap and significantly improves computational efficiency. Our method requires $16.7\times$ fewer network updates than a jointly-trained (upper bound) model. In terms of TFLOPs, our method provides a $31.9\times$ speedup (see Fig. 5). For the IID CL distribution, the method achieves backward transfer, where learning the new dataset helps to improve ImageNet-1K accuracy.

4. We show that our method is effective and we integrate it into existing CL methods. Specifically, it performs well in conjunction with the rehearsal methods Vanilla Rehearsal, DERpp, GDumb, and REMIND as well as the non-rehearsal method LwF.

## 2 Background

Many methods have been proposed to learn continuously from non-stationary datasets in continual learning (see Zhou et al. (2023) for review). These methods can be broadly divided into three categories: 1) **Rehearsal-based methods** store or reconstruct a subset of old data to rehearse alongside new data while learning a new batch (Chaudhry et al., 2019; Hou et al., 2019; Rebuffi et al., 2017; Wu et al., 2019), 2) **Regularization-based methods** constrain weight updates by adding additional regularization in the loss function (Aljundi et al., 2018; Chaudhry et al., 2018; Dhar et al., 2019; Kirkpatrick et al., 2017), and 3) **Parameter-isolation based methods** allocate multiple sets of parameters or multiple copies of the model to different incremental batches (Douillard et al., 2021; Yan et al., 2021; Yoon et al., 2020). Since parameter

---

[2]Referring to the loss when we only train the output layer (i.e., final classification layer) and freeze all other layers.

isolation methods do not allow backward transfer, the stability gap does not apply to them De Lange et al. (2023).

While most of the CL research community has focused on mitigating catastrophic forgetting (McCloskey & Cohen, 1989), there is a growing body of literature demonstrating its ability to reduce the amount of compute needed to update the network (Ghunaim et al., 2023; Harun et al., 2023a;b;c; Prabhu et al., 2023b; Verwimp et al., 2024). Recently, a major obstacle to this objective has been identified in CL: *The stability gap* (De Lange et al., 2023). However, it has been studied on small datasets, e.g., CIFAR, using randomly initialized DNNs. This setting is not aligned with studying model decay in industrial settings that involve many-class data streams and preserving performance in large pre-trained DNNs.

Traditionally, the *training-from-scratch* CL would start with a randomly initialized DNN that had no pre-training. With the prevalence of large pre-trained models, recent CL research has started integrating these models into the learning process. This has resulted in a growing number of CL works that utilize large pre-trained vision transformer (ViT) models, which have been pre-trained on ImageNet-1K or ImageNet-21K (Wang et al., 2022b;a; Smith et al., 2023; Gao et al., 2023; McDonnell et al., 2024). Moreover, many earlier works use pre-training for CL (Belouadah & Popescu, 2019; Hou et al., 2019; Castro et al., 2018; Rebuffi et al., 2017; Hayes et al., 2020; Douillard et al., 2020; Harun et al., 2023b) and emphasize the significance of pre-training in the context of CL (Lee et al., 2023; Mehta et al., 2023; Ostapenko et al., 2022; Ramasesh et al., 2021b; Gallardo et al., 2021). However, as demonstrated in several works (Wang et al., 2022a; Mirzadeh et al., 2022), using pre-trained models does not naively enhance CL performance and effectively leveraging pre-trained models for CL remains an open question. In this work, we mainly focus on overcoming the stability gap in CL with pre-trained models.

Almost all CL methods focus on the catastrophic forgetting problem with evaluations occurring on discrete batch or task transitions (Hayes et al., 2018; Chaudhry et al., 2018) and fail to capture the stability gap that occurs immediately after a new task is introduced (Fig. 1). De Lange et al. (2023) demonstrates the stability gap occurs for a variety of CL methods, including rehearsal (Chaudhry et al., 2019), GEM (Lopez-Paz & Ranzato, 2017), EWC (Kirkpatrick et al., 2017)), and LwF (Li & Hoiem, 2017). To quantify the stability gap, they propose continual evaluation metrics based on worst-case performance i.e., the largest drop in accuracy on old batches. However, their metrics do not enable comparing *different* CL models because they only quantify the largest drop relative to the *same* model's best performance, thus making it impossible to quantify the impact of different training methodologies aimed at mitigating the stability gap. We address this limitation using novel metrics that are normalized against a jointly-trained universal upper bound (see Sec. 3.2). We also illustrate this phenomenon using synthetic examples in Appendix F to build intuitions. We study overcoming the stability gap with both rehearsal and non-rehearsal methods, with a focus on rehearsal due to its effectiveness (van de Ven et al., 2022; Zhou et al., 2023).

## 3 Continual Learning Protocol

In this work, we attempt to align CL with the industry setting, where CL updates knowledge in pre-trained models to prevent model decay when new concepts are introduced while preserving performance on the original classes. Likewise, because the industry requires maintaining high accuracy, we focus on rehearsal-based CL; nonetheless, we demonstrate that our mitigation strategies can be used for non-rehearsal methods that use regularization and knowledge distillation. In this section, we formalize our CL framework and define the metrics we use to measure the stability gap.

### 3.1 Formal Continual Learning Setting

To align our work with addressing model decay, we assume CL begins with a pre-trained model capable of $K$-way classification. The goal is to incorporate new data, which may have additional classes, into the model, while preserving or improving performance on the original $K$ classes. For this purpose, we use ImageNet-1K pre-trained models ($K = 1000$). While ImageNet-1K pre-trained models are commonly employed in CL systems, previous studies do not attempt to maintain or measure performance on ImageNet-1K dataset itself (Wang et al., 2022b;a; Smith et al., 2023; Gao et al., 2023; McDonnell et al., 2024). Our research

uniquely explores this aspect in CL. Upon deployment, the pre-trained model is exposed to a sequence of data batches over $N - 1$ learning sessions, i.e., $\{\mathcal{S}_2, \cdots, \mathcal{S}_N\}$, where $\mathcal{S}_1$ is the data used for pre-training. The $j$'th session consists of a batch of $m_j$ labeled training samples, i.e., $\mathcal{S}_j = \{(\mathbf{x}_j, y_j)_i\}_{i=1}^{m_j}$, where $\mathbf{x}_j$ is an instance of category $y_j \in Y_j$ and $Y_j$ is the label space of task $j$. In the CIL setting, each $\mathcal{S}_j$ contains non-overlapping classes, i.e., $Y_j \cap Y_{j'} = \emptyset$ for $j \neq j'$. The total number of samples in the entire sequence is $M = \sum_{j=1}^{N} m_j$. When learning new batch $\mathcal{S}_j$, the learner can access $\mathcal{S}_j$ and any stored data from previous batches $\mathcal{S}_{1:j-1}$. During test time, the learner is evaluated on test data from all seen classes, $\mathcal{Y}_j = Y_1 \cup \cdots Y_j$. The batch (or task) identifier $j$ cannot be exploited during test time.

To efficiently adapt to a large-scale data stream in a real-world setting, a CL system should not increase compute cost over time. For all learning sessions $j$ after pre-training, it is given a fixed compute budget $\mathcal{B}$ corresponding to the number of SGD model updates i.e., $\mathcal{B} = \mathcal{U} \times b$, where $\mathcal{U}$ and $b$ denote the number of training iterations and the number of training samples per iteration respectively.

### 3.2 Measuring the Stability Gap

Popular CL metrics focus on measuring performance after each $\mathcal{S}_j$ is learned. They do not permit fine-grained analysis for a) preserving old knowledge, b) acquiring new knowledge and c) balancing both during training. To study the stability gap in a manner that enables us to test our hypotheses across models trained with different strategies, we created metrics that measure these criteria: 1) the stability gap, $\mathcal{S}_\Delta$ (criterion a) 2) the plasticity gap, $\mathcal{P}_\Delta$ (criterion b), and 3) the continual knowledge gap, $\mathcal{CK}_\Delta$ (criterion c). Each metric asks: ***How much performance does a learner lack on previously observed, recently observed, or all observed data compared to a joint model (upper bound) when learning new data?***

For the $j$'th learning session, we denote evaluation sets on old, new, and all seen data by $\mathrm{E}_{1:j-1}$, $\mathrm{E}_j$, and $\mathrm{E}_{1:j}$, respectively. $\mathcal{A}_i$ is the accuracy of the current model $\theta_i$ on batch evaluation set E at training iteration $i$, and $L$ is the total number of iterations. $\mathcal{A}_{joint}$ is the final accuracy of a joint model ($\theta_{joint}$) trained jointly on all data. We define the **stability gap** as

$$\mathcal{S}_\Delta = 1 - \frac{1}{N-1} \sum_{j=2}^{N} \Omega_j^{old}; \text{ where } \Omega_j^{old} = \frac{1}{L} \sum_{i=1}^{L} \frac{\mathcal{A}_i(\mathrm{E}_{1:j-1}, \theta_i)}{\mathcal{A}_{joint}(\mathrm{E}_{1:j-1}, \theta_{joint})}. \tag{1}$$

Similarly, the **plasticity gap** is

$$\mathcal{P}_\Delta = 1 - \frac{1}{N-1} \sum_{j=2}^{N} \Omega_j^{new}; \text{ where } \Omega_j^{new} = \frac{1}{L} \sum_{i=1}^{L} \frac{\mathcal{A}_i(\mathrm{E}_j, \theta_i)}{\mathcal{A}_{best}(\mathrm{E}_j, \theta_{best})}, \tag{2}$$

where $\mathcal{A}_{best}$ stands for the best accuracy achieved by the best CL model ($\theta_{best}$) at any time during training. And finally, we define the **continual knowledge gap** as

$$\mathcal{CK}_\Delta = 1 - \frac{1}{N-1} \sum_{j=2}^{N} \Omega_j^{all}; \text{ where } \Omega_j^{all} = \frac{1}{L} \sum_{i=1}^{L} \frac{\mathcal{A}_i(\mathrm{E}_{1:j}, \theta_i)}{\mathcal{A}_{joint}(\mathrm{E}_{1:j}, \theta_{joint})}. \tag{3}$$

$\Omega_j$ records CL performance compared to $\mathcal{A}_{joint}$ or $\mathcal{A}_{best}$. After learning all $N$ batches, $\Omega_j$ scores are averaged to indicate average performance gain. The first batch is excluded since that is used for pre-training. For all metrics, smaller $\mathcal{S}_\Delta$, $\mathcal{P}_\Delta$, and $\mathcal{CK}_\Delta$ indicate better performance. When $\mathcal{A}_i = \mathcal{A}_{joint}$ and $\mathcal{A}_i = \mathcal{A}_{best}$ for all $L$ iterations, $\mathcal{S}_\Delta$, $\mathcal{P}_\Delta$, and $\mathcal{CK}_\Delta$ become zero which is desirable. A negative value means knowledge transfer between new and old batches, which is desirable. These metrics apply for offline CL (i.e., incremental-batch CL) and online CL with any data distributions, including CIL and IID.

## 4 Stability Gap Mitigation Methods

This section describes the methods we use to test our hypotheses regarding the factors that increase the stability gap. We use our evaluation metrics, $\mathcal{S}_\Delta$, $\mathcal{P}_\Delta$, and $\mathcal{CK}_\Delta$ to validate the efficacy of proposed mitigation methods. We include additional discussion in Appendix H.

**Weight Initialization**. In CIL, the output units for new classes are typically randomly initialized causing those units to produce a high loss during backpropagation. We hypothesize that using data-driven initialization for new class units will reduce the loss and therefore reduce the stability gap, $\mathcal{S}_\Delta$. To test this, we initialize them to the mean of unit length embeddings for that class, i.e.,

$$\mathbf{w}_k = \frac{1}{V} \sum_{j=1}^{V} \frac{\mathbf{h}_j}{\|\mathbf{h}_j\|_2}, \tag{4}$$

where $\mathbf{w}_k \in \mathbb{R}^d$ is the output layer weight vector for class $k$, $\mathbf{h}_j \in \mathbb{R}^d$ is the $j$'th embedding from the penultimate layer, and $V$ is the number of samples from class $k$ in the batch. Intuitively, this approach aims to align the initial weights more closely with the distribution of the new class's data, facilitating faster and more effective learning adjustments during the initial phase of CL.

**Hard vs. Dynamic Soft Targets**. For classification, models are often trained with hard targets, i.e., at training iteration $i$ a one-hot vector $\mathbf{t}_i$ with a '1' in position $k$ corresponding to the correct class. We hypothesize hard target training is partially responsible for the stability gap. Intuitively, hard targets are one-hot encoded and enforce strict inter-class independence despite several classes sharing distributional similarities. This property of hard targets, therefore, also causes a large initial loss when learning new classes. Soft targets, on the other hand, can help the network retain the joint inter-class distributions, which further ameliorates the perturbation of learned classes. To test this, we use soft targets constructed such that the model's predictions on previously learned classes are largely preserved.

At learning iteration $i$, let $P(k|\mathbf{x}_i; \theta_i)$ be the model's output softmax probabilities for sample $\mathbf{x}_i$ from class $k$ given the model's current parameters $\theta_i$ and the predicted class be $y_i' = \arg\max_k P(k|\mathbf{x}_i; \theta_i)$. We maintain a running average vector $\mathbf{u}_k \in \mathbb{R}^K$ of the softmax probabilities for each class that is updated when an example from class $k$ is observed, i.e.,

$$\mathbf{u}_k \leftarrow \frac{c_k \mathbf{u}_k + P(k|\mathbf{x}_i; \theta_i)}{c_k + 1}, \tag{5}$$

where $c_k$ is a counter for class $k$ that is subsequently increased by 1, and $\mathbf{u}_k$ is initialized to a uniform distribution prior to the running updates. Subsequently, soft targets $\mathbf{t}_i$ for iteration $i$ are constructed by setting $\mathbf{t}_i \leftarrow \mathbf{u}_k$ and then setting the element for the correct class to 1, i.e., $\mathbf{t}_i[k] \leftarrow 1$. If $y_i' \neq k$, then we also set $\mathbf{t}_i[y_i'] \leftarrow 1/K$. Subsequently, $\mathbf{t}_i$ is normalized to sum to 1 and used to update the network. This strategy results in targets that minimally perturb the network and smaller loss values. In Appendix. G, we illustrate the process of updating soft targets.

**Limiting Hidden Layer Plasticity Using LoRA**. To accumulate knowledge over time, most CL approaches update the entire network. Given that each batch of data in CL is relatively small, we hypothesize that this leads to excessively perturbing hidden representations, leading to a larger stability gap. To test this hypothesis, we constrain the number of trainable parameters in hidden representations using a network adaptor. While there are various network adaptors (Han et al., 2024) that restrict network plasticity, we use low-rank adaptation (LoRA) (Hu et al., 2021) due to its simplicity and effectiveness. Specifically, we inject LoRA weights into the linear layers of the network, and only these parameters and the output layer are updated, which greatly reduces the number of trainable parameters.

For batch $j$, let $\mathbf{W}^{j-1} \in \mathbb{R}^{d \times g}$ be a previously learned linear layer. At the start of each learning session, we reparameterize this layer by replacing $\mathbf{W}^{j-1}$ with

$$\mathbf{\Theta}^j = \mathbf{W}^{j-1} + \mathbf{B}\mathbf{A}, \tag{6}$$

where $\mathbf{B} \in \mathbb{R}^{d \times r}$ and $\mathbf{A} \in \mathbb{R}^{r \times g}$ are the LoRA adapter parameters with rank $r \ll \min(d, g)$. Only $\mathbf{B}$ and $\mathbf{A}$ are plastic, with $\mathbf{A}$ initialized with random Gaussian values and $\mathbf{B}$ initialized to a zero matrix, so $\mathbf{B}\mathbf{A} = \mathbf{0}$ at the beginning of the learning session. At the end of the session, the LoRA parameters are folded into the network, i.e., $\mathbf{W}^j \leftarrow \mathbf{\Theta}^j$. In LoRA experiments, only the output layer and the LoRA parameters are plastic.

**Limiting Output Layer Plasticity via Targeted Freezing**. In CIL, large changes in the network's representations for old classes increase the stability gap. While LoRA restricts plasticity in hidden representations, we hypothesize that restricting plasticity in the output layer could also be helpful for CIL. Therefore,

we study freezing output layer weights for classes previously learned in earlier batches. For rehearsal methods, samples from classes seen in earlier batches have the hidden layers updated as usual. We refer to this technique as old output class freezing (OOCF).

**Combining Mitigation Methods & SGM**. We independently evaluate each of the stability gap mitigation methods. Additionally, we evaluate them in combination. We refer to the method that combines dynamic soft targets, weight initialization, OOCF, and LoRA as SGM (**S**tability **G**ap **M**itigation). Soft targets and weight initialization prevent higher loss at the output layer to enhance stability. OOCF and LoRA restrict plasticity in the network to targeted locations so that existing representations are minimally perturbed.

## 5 Main Experiments: Hypothesis Evaluation

We design our experiments to emulate using CL to mitigate model decay. Hence, we use an ImageNet-1K pre-trained DNN that is progressively updated with new data and classes. As shown in Fig. 1 and Fig. 4, the stability gap is present when vanilla rehearsal is employed.

### 5.1 Experimental Setup

**Bounding Compute.** In a real-world scenario, a continual learner must adapt to a large-scale data stream, which may not be feasible if more computation is required over time. Recently, many works have advocated computational efficiency for CL (Prabhu et al., 2023b; Harun et al., 2023b;a;c; Verwimp et al., 2024; Zhang et al., 2023). In our experiments, we bound compute by a fixed number of training iterations or SGD steps.

**Rehearsal.** Because our goal is to understand and mitigate the stability gap, our main results use rehearsal and assume the learner has access to all previously observed data, with no constraints on storage. This is aligned with finding a better alternative to periodically retraining from scratch as more data is acquired, which is commonly done in industry where the computational budget depends on compute to a far greater extent than data storage (Prabhu et al., 2023b;a).

**Continual Learning Procedure.** We aim to study CL in an industry-like setting, requiring a large-scale data stream with numerous object categories. However, it is difficult to find a suitable dataset well-curated and suitable for large-scale CL experiments. Therefore, we construct a large-scale data stream by combining ImageNet-1K with another dataset, Places365-LT, which is a variant of the Places365 dataset. Places365 is challenging (Liu et al., 2019) and is widely used for out-of-distribution (OOD) detection with ImageNet pre-trained models (Zhu et al., 2022).

During CL, the model sequentially learns 5 incremental batches of data from Places365-LT. In the CIL ordering, each CL batch contains 73 categories, and in the IID ordering each CL batch has 12500 examples. During rehearsal, the model is updated over 600 minibatches, where the minibatch consists of 128 samples where 50% are selected randomly from the current CL batch and 50% from data seen in earlier CL batches and ImageNet-1K. We study two CL orderings for Places365-LT: 1) CIL ordering where each batch has classes exclusively seen in that batch (maximum concept drift), and 2) IID ordering where each batch contains examples from randomly sampled classes (minimal concept drift). These are two opposite extreme situations in CL. Although the stability gap is not known to occur in the IID CL setting, this setting is critical to demonstrating the algorithm's generality. To measure the stability gap and other metrics, we assess performance during rehearsal every 50 training iterations, where the test set consists of the ImageNet-1K validation set and all of the classes from Places365-LT from the current and prior CL batches. Additional implementation and dataset details are given in Appendix C and B.

**Network Architecture.** We choose a network architecture that is amenable to LoRA and performs well in offline training with ImageNet-1K compared to similar-sized DNNs. In our main results, we study CL using the **ConvNeXtV2-Femto** (Woo et al., 2023) CNN that has been pre-trained on ImageNet-1K using a fully convolutional masked autoencoder framework followed by supervised fine-tuning on ImageNet-1K. While ResNet18 is widely used in CL, it under-performs other lightweight CNNs in CL (Hayes & Kanan, 2022; Harun et al., 2023b). ConvNeXtV2-Femto has 5.2M parameters, which is 2× less than ResNet18's 11.6M parameters, and it outperforms ResNet18 by absolute 8.47% in final top-1 accuracy on ImageNet-1K.

Table 1: **Hypothesis Evaluation (CIL).** Results after learning ImageNet-1K followed by Places365-LT over 5 rehearsal sessions. $\mu$ denotes average accuracy (%) over rehearsal sessions and $\alpha$ is final accuracy (%) on all 1365 classes. $\sigma$ stands for final accuracy (%) on ImageNet-1K only. $\#P$ denotes the trainable parameters in Millions. The best and 2nd best values are indicated in bold and underlined respectively.

| Method | $\#P\downarrow$ | $\mathcal{S}_\Delta\downarrow$ | $\mathcal{P}_\Delta\downarrow$ | $\mathcal{CK}_\Delta\downarrow$ | $\sigma\uparrow$ | $\mu\uparrow$ | $\alpha\uparrow$ |
|---|---|---|---|---|---|---|---|
| Joint Model (Upper Bound) | 5.08 | — | — | — | 77.58 | — | 70.69 |
| Naive Finetune (Lower Bound) | 5.08 | 0.743 | 0.474 | 0.739 | 16.77 | 14.03 | 15.60 |
| Output Layer Only | **0.53** | 0.026 | 0.494 | 0.032 | 76.10 | 71.23 | 67.89 |
| Vanilla Rehearsal | 5.08 | 0.028 | 0.393 | 0.033 | 76.06 | 71.52 | 67.67 |
| Dynamic Soft Targets | 5.08 | 0.022 | 0.397 | 0.029 | 76.26 | 71.78 | 68.24 |
| Weight Initialization | 5.08 | 0.024 | 0.097 | 0.020 | 76.69 | 72.43 | 69.22 |
| OOCF | 5.08 | 0.026 | 0.376 | 0.032 | 75.95 | 71.57 | 67.94 |
| LoRA | 1.45 | 0.018 | 0.316 | 0.019 | 76.92 | 72.74 | 69.19 |
| **SGM** (Combined Method) | 1.45 | **0.006** | **0.082** | **0.002** | **77.64** | **73.70** | **70.30** |

It is worth noting that LoRA is not appropriate for CNN architectures (e.g., ResNet) that lack $1 \times 1$ convolutional layers, but it can be used with ViT, ConvNeXtV1, ConvNeXtV2, and other DNN architectures with these linear layers. Each block of ConvNeXt consists of one 2D convolutional layer and two $1 \times 1$ convolutional layers. For experiments using LoRA, the $1 \times 1$ convolutional layers in ConvNeXt blocks are modified to incorporate LoRA's weights using Equation 6. The number of trainable LoRA weights is 0.92M, which is much less than the total number of hidden layer parameters (5.08M). Based on prior work, early layers in the network are universal feature extractors and are little altered during CL (Ramasesh et al., 2021a; Ebrahimi et al., 2020; Pellegrini et al., 2020; Harun et al., 2024), so we freeze the first 4 blocks of the CNN in all experiments, leaving the remaining 8 blocks plastic (97.7% of the parameters) which consist of 5.08M parameters.

## 5.2 Experimental Results

We describe our findings on how the proposed method (SGM) mitigates the stability gap and enhances learning efficiency in an unlimited storage setting with rehearsal in CIL.

### 5.2.1 Evaluating Our Hypotheses

Here, we describe the results for evaluating our two hypotheses.

**Baselines.** In our experiments, we evaluate each of the methods in Sec. 4 individually and the combined SGM method. As baselines, we compare SGM with rehearsal against vanilla rehearsal, which uses unlimited storage for rehearsal without any additional components, as well as joint models (upper bound). The joint models are jointly trained on ImageNet and the CL batches seen up to the current batch. We also include naive finetuning which is a vanilla variant without rehearsal and serves as a lower bound. Finally, we compare with the output layer only which updates the output layer with rehearsal while freezing all other layers.

**Results.** Our main CIL results are given in Table 1. SGM with rehearsal shows the greatest reduction in the stability gap ($\mathcal{S}_\Delta$), plasticity gap ($\mathcal{P}_\Delta$), and continual knowledge gap ($\mathcal{CK}_\Delta$). It also performs best in other metrics. Of its components, LoRA reduces the stability gap the most; however, the stability gap for LoRA is 3× higher than SGM. We next turn to examining the support for our two hypotheses.

**Hypothesis 1.** Our first hypothesis was that the stability gap in CIL is caused by having a large loss at the output layer due to the new classes, which we tested by using weight initialization and dynamic soft targets. Both methods are effective at achieving the goal of reducing the initial loss, especially weight initialization (see Fig. 2a). As shown in Table 1, both methods reduce the stability gap. We observe that weight initialization also greatly reduces the plasticity gap. Fig. 2b shows the average performance on ImageNet-1K during the 5 rehearsal sessions for hard vs. soft targets, which reveals that hard targets increase the stability gap to a greater extent than dynamic soft targets.

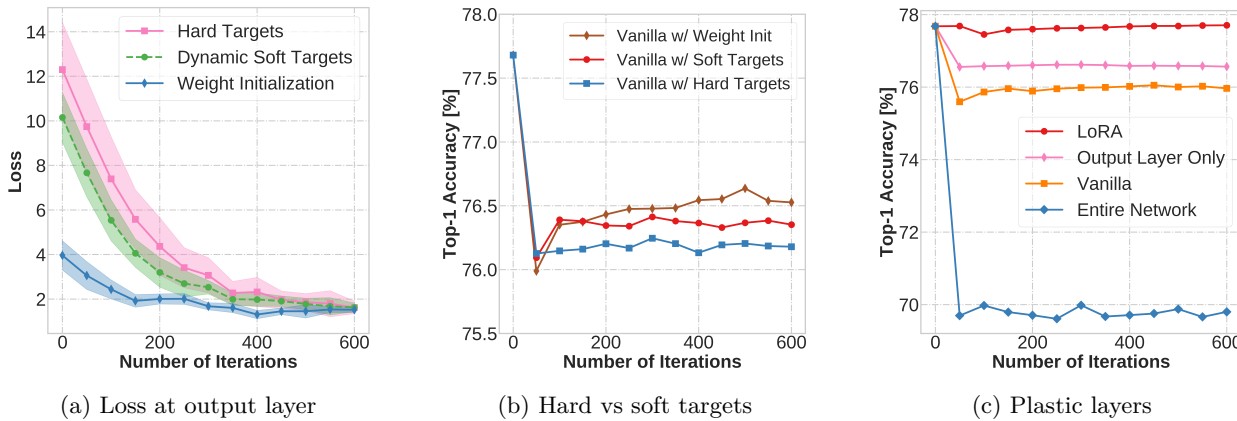

(a) Loss at output layer      (b) Hard vs soft targets      (c) Plastic layers

Figure 2: Mitigation methods averaged over 5 rehearsal sessions during CIL. (a) The loss on new classes when only training the output layer, which reveals soft targets and data-driven weight initialization greatly reduce the initial loss. (b) Accuracy on ImageNet-1K for hard vs. soft targets, which shows that soft targets reduce the stability gap. (c) Network plasticity increases the stability gap.

Table 2: **Average CIL Results.** Results after learning ImageNet-1K followed by Places365-LT over 5 rehearsal sessions. $\mu$ denotes average accuracy (%) over rehearsal sessions and $\alpha$ is final accuracy (%) on all 1365 classes. $\sigma$ stands for final accuracy (%) on ImageNet-1K only. $\#P$ denotes the trainable parameters in Millions. We report the average over 6 data orderings with standard deviation ($\pm$).

| Method | $\#P \downarrow$ | $\mathcal{S}_\Delta \downarrow$ | $\mathcal{P}_\Delta \downarrow$ | $\mathcal{CK}_\Delta \downarrow$ | $\sigma \uparrow$ | $\mu \uparrow$ | $\alpha \uparrow$ |
|---|---|---|---|---|---|---|---|
| Joint | 5.08 | — | — | — | 77.58 | — | 70.69 |
| Vanilla | 5.08 | 0.020 ±0.0017 | 0.385 ±0.0091 | 0.031 ±0.0015 | 75.81 ±0.1585 | 71.68 ±0.1236 | 67.94 ±0.1721 |
| Output Layer | **0.53** | 0.021 ±0.0011 | 0.473 ±0.0250 | 0.032 ±0.0009 | 75.79 ±0.2875 | 71.28 ±0.1410 | 67.68 ±0.2710 |
| **SGM** | 1.45 | **0.001** ±0.0012 | **0.087** ±0.0082 | **0.002** ±0.0007 | **77.70** ±0.0509 | **73.71** ±0.0763 | **70.31** ±0.0682 |

**Hypothesis 2.** Our second hypothesis was that the stability gap in CIL is caused by excessive network plasticity, which we tested by using LoRA and OOCF. Fig. 2c shows the average results on ImageNet-1K across the 5 rehearsal sessions for LoRA vs. when only the output layer, top 8 blocks (vanilla), or entire network are trainable. This reveals that plasticity plays a major role in the stability gap; however, this does not translate directly into the number of trainable parameters since LoRA includes the output layer but exhibits a smaller decrease in performance than training only the output layer. Unlike others, LoRA fully recovers old performance. As seen in Table 1, both OOCF and especially LoRA reduce the stability gap.

**Interim Conclusions.** Our experimental results support both hypotheses. LoRA mitigates the stability and continual knowledge gaps the most. Our data-driven weight initialization greatly improves the plasticity. This aligns with the prior study that suggests that weight initialization is critical for plasticity in DNN (Lyle et al., 2023). We find that SGM, a method that combines the approaches used to test these hypotheses, greatly reduces the stability gap.

**CIL with Multiple Data Orderings.** To ensure the robustness of our findings, we extend our experiments to multiple data orderings. In addition to studying SGM and vanilla rehearsal, we examine only training the output layer during rehearsal. The averaged results across 6 data orderings are given in Table 2. We find that SGM consistently mitigates the stability gap. Compared to vanilla, SGM provides 20× more stability, 4.4× more plasticity, and 15.5× more continual knowledge. SGM also outperforms training only the output layer in all criteria. This indicates that updating representations in hidden layers besides the output layer using SGM is critical for learning new and retaining old knowledge. Moreover, SGM balances stability and plasticity (see Fig. 6).

### 5.2.2 Learning Efficiency

One of our goals in studying CL and the stability gap is to enable more computationally efficient training. To study whether SGM achieves this goal, we evaluated performance during CIL, where we measured performance on old and new classes every 10 iterations. For new classes, we measured the number of updates and FLOPs (floating-point operations) needed to achieve 99% of the best accuracy.

As shown in Fig. 3, SGM learns new classes much faster than vanilla rehearsal, and on average, it is 61.8% more efficient in terms of network updates. Moreover, because the curve shows a decreasing trend as learning progresses this means that SGM becomes a more efficient learner over time, with fewer updates and TFLOPs for the current batch than previous batches being needed. For old class performance, we measured the number of iterations during rehearsal required to recover the performance of a joint model (upper bound) on ImageNet-1K for vanilla and SGM. As shown in Table 3, SGM requires far fewer iterations to recover old performance whereas vanilla rehearsal (without SGM) fails to recover old performance fully. Compared to a joint model, SGM provides a 16.7× speedup in number of network updates and a 31.9× speedup in TFLOPs (see Fig. 5). Moreover, SGM requires 18× less training time than the joint model on the same hardware.

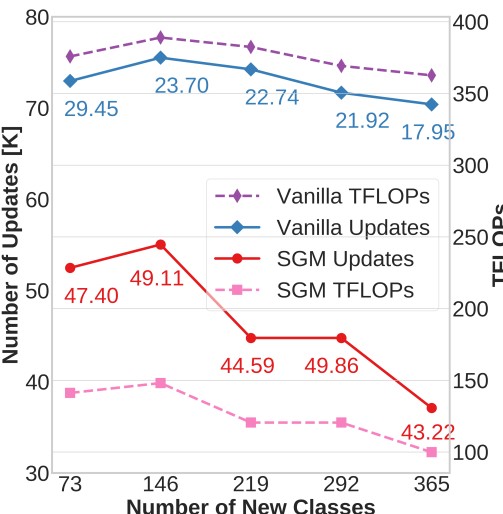

Figure 3: **Speed of acquiring new knowledge.** SGM requires fewer updates and TFLOPs than vanilla to reach 99% of the best accuracy on new classes (highlighted).

Table 3: **Speed of Recovering Old Knowledge.** SGM compared to vanilla (without SGM) for the number of iterations needed to recover 97%, 98%, 99%, or 100% of the accuracy on ImageNet-1K for a joint model as each of the 5 tasks (denoted by $T_i$ where $i \in \{1, 5\}$) from Places365-LT is learned during CIL. A hyphen indicates the model did not recover performance whereas zero means there was no stability gap.

| Recovery | With SGM | | | | | Without SGM | | | | |
|---|---|---|---|---|---|---|---|---|---|---|
| | $T_1$ | $T_2$ | $T_3$ | $T_4$ | $T_5$ | $T_1$ | $T_2$ | $T_3$ | $T_4$ | $T_5$ |
| 100% | 260 | 0 | 70 | 50 | 80 | — | — | — | — | — |
| 99% | 110 | 0 | 70 | 0 | 70 | — | — | — | — | — |
| 98% | 90 | 0 | 70 | 0 | 70 | 60 | 450 | 110 | — | — |
| 97% | 90 | 0 | 70 | 0 | 70 | 30 | 450 | 10 | 360 | 10 |

## 6 Additional Experiments: SGM's Generality

This section expands on the versatility and efficacy of the SGM method by exploring its application across various settings and constraints. First, we show how SGM performs in the IID CL setting in Sec. 6.1. Next, we study storage-constrained rehearsal in both offline CL (Sec. 6.2) and online CL (Sec. 6.3) scenarios with CIL data ordering. Finally, we summarize additional supporting studies in Sec. 6.4. Additional implementation details and dataset details are given in Appendix C and B.

### 6.1 IID Continual Learning

To understand if SGM with rehearsal would be useful for other CL data distributions, we examine its behavior in an IID ordering where each of the 5 CL batches contains randomly sampled classes from **Places365-LT** dataset. In this experiment, we use **ConvNeXtV2-Femto** pre-trained on ImageNet-1K using self-supervised learning (SSL). During IID CL, the model sequentially learns 5 incremental batches of data from Places365-LT where each incremental batch contains 12500 examples. During rehearsal, the model is updated over 600

Table 4: **IID CL Results**. Results after learning ImageNet-1K followed by Places365-LT over 5 rehearsal sessions. $\mu$ denotes average accuracy (%) over rehearsal sessions and $\alpha$ is final accuracy (%) on all 1365 classes. $\sigma$ stands for final accuracy (%) on ImageNet-1K only. $\#P$ denotes trainable parameters in Millions. We report the average over 5 data orderings with standard deviation ($\pm$).

| Method | $\#P \downarrow$ | $\mathcal{S}_\Delta \downarrow$ | $\mathcal{P}_\Delta \downarrow$ | $\mathcal{CK}_\Delta \downarrow$ | $\sigma \uparrow$ | $\mu \uparrow$ | $\alpha \uparrow$ |
|---|---|---|---|---|---|---|---|
| Joint | 5.08 | — | — | — | 77.58 | – | 70.69 |
| Vanilla | 5.08 | 0.014 ±0.0004 | 0.173 ±0.0056 | 0.034 ±0.0005 | 76.12 ±0.1007 | 68.45 ±0.0470 | 68.77 ±0.0517 |
| **SGM** | **1.45** | **−0.004** ±0.0005 | **0.131** ±0.0017 | **0.003** ±0.0004 | **77.84** ±0.0234 | **70.81** ±0.0151 | **71.23** ±0.0664 |

Table 5: **Offline CL.** SGM's generality results for CIL on a combination of ImageNet-1K and Places365-Standard. SGM† and SGM‡ denote variants of DERpp and GDumb respectively when SGM is integrated with them. $\mu$ denotes average accuracy (%) over rehearsal sessions and $\alpha$ is final accuracy (%) on all 1365 classes. $\#P$ denotes the trainable parameters in Millions.

| Method | $\#P \downarrow$ | Storage Constraint 192K Samples | | | | | Storage Constraint 24K Samples | | | | |
|---|---|---|---|---|---|---|---|---|---|---|---|
| | | $\mathcal{S}_\Delta \downarrow$ | $\mathcal{P}_\Delta \downarrow$ | $\mathcal{CK}_\Delta \downarrow$ | $\mu \uparrow$ | $\alpha \uparrow$ | $\mathcal{S}_\Delta \downarrow$ | $\mathcal{P}_\Delta \downarrow$ | $\mathcal{CK}_\Delta \downarrow$ | $\mu \uparrow$ | $\alpha \uparrow$ |
| Joint | 5.08 | — | — | — | — | 65.37 | — | — | — | — | 65.37 |
| DERpp | 5.08 | 0.142 | 0.109 | 0.126 | 62.53 | 53.38 | 0.209 | 0.109 | 0.187 | 57.25 | 44.74 |
| **SGM†** | **1.45** | **0.071** | **0.091** | **0.061** | **67.41** | **57.28** | **0.086** | **0.095** | **0.074** | **66.29** | **55.98** |
| GDumb | 5.08 | 0.133 | 0.110 | 0.120 | 62.85 | 54.72 | 0.224 | 0.114 | 0.202 | 55.54 | 43.10 |
| **SGM‡** | **1.45** | **0.053** | **0.095** | **0.046** | **68.50** | **59.24** | **0.073** | **0.098** | **0.065** | **66.88** | **57.18** |

minibatches, where the minibatch consists of 128 samples with 50% randomly selected from the current CL batch and 50% drawn from data seen in earlier CL batches and ImageNet-1K.

In this experiment, we omit storage constraints to solely focus on mitigation methods without the influence of other variables. Our results are summarized in Table 4. In terms of final accuracy, SGM achieves a final accuracy of 71.23%, outperforming vanilla rehearsal's 68.77% accuracy. Surprisingly, SGM even surpasses the jointly trained model's 70.69% accuracy, resulting in a *negative* stability gap, which indicates knowledge transfer from new classes to old classes. In contrast, we found there was a small stability gap in CIL ordering (see Table 2), likely due to the dissimilarity among subsequent batches.

## 6.2 Storage Constrained Offline Continual Learning

To study SGM's efficacy under storage constraints, we combine it with two popular rehearsal methods, DERpp (Buzzega et al., 2020) and GDumb (Prabhu et al., 2020), under varied storage constraints while using identical configurations. For this, we use **ConvNeXtV2-Femto** pre-trained on ImageNet-1K using SSL. To study varied storage constraints on a large-scale data stream, we combine the ImageNet-1K (1.2M images) dataset with another large-scale dataset, **Places365-Standard** (1.8M images). This allows us to test more restrictive storage constraints on a data stream consisting of 3 million images.

After being pre-trained on ImageNet-1K, a model sequentially learns 5 incremental batches of data (73 classes per batch) from Places365-Standard in offline CL with CIL ordering. Each rehearsal is compute-constrained where the model is updated over 1200 minibatches. Each minibatch consists of 256 samples where 50% are selected randomly from the current CL batch and 50% from data seen in earlier CL batches and ImageNet-1K. During learning a new batch, the model rehearses old data from storage which is bounded by a maximum number of samples e.g., 192K and 24K corresponding to 6.4% and 0.8% of the entire dataset (ImageNet and Places combined), respectively. As shown in Table 5, SGM improves each method's performance in all criteria. When the storage is bounded by 24K samples, SGM improves final accuracy by absolute 11.24% (DERpp) and 14.08% (GDumb) and provides 2.4× and 3.1× more stability for DERpp and GDumb, respectively.

Table 6: **Online CL.** Results when SGM is combined with REMIND (denoted by SGM†) for CIL on ImageNet-1K and CUB-200. $\mu$ denotes average accuracy (%) over rehearsal sessions and $\alpha$ is final accuracy (%) on all 1200 classes. $\#P$ denotes trainable parameters in Millions.

| Method | $\#P \downarrow$ | Storage Constraint 80K Samples | | | | | Storage Constraint 20K Samples | | | | |
|---|---|---|---|---|---|---|---|---|---|---|---|
| | | $\mathcal{S}_\Delta \downarrow$ | $\mathcal{P}_\Delta \downarrow$ | $\mathcal{CK}_\Delta \downarrow$ | $\mu \uparrow$ | $\alpha \uparrow$ | $\mathcal{S}_\Delta \downarrow$ | $\mathcal{P}_\Delta \downarrow$ | $\mathcal{CK}_\Delta \downarrow$ | $\mu \uparrow$ | $\alpha \uparrow$ |
| Joint | 5.08 | — | — | — | — | 75.99 | — | — | — | — | 75.99 |
| REMIND | 5.08 | 0.146 | 0.837 | 0.162 | 64.15 | 62.35 | 0.166 | 0.844 | 0.183 | 62.58 | 60.37 |
| **SGM†** | **1.45** | **0.034** | **0.675** | **0.049** | **72.81** | **69.67** | **0.042** | **0.695** | **0.057** | **72.20** | **69.01** |

## 6.3 Storage Constrained Online Continual Learning

We also assess SGM's efficacy in an online CL setting using a state-of-the-art online CL method, RE-MIND (Hayes et al., 2020). Using **ConvNeXtV2-Femto**, we conduct storage-constrained CL experiments with the CIL data ordering, where we combine SGM with REMIND. Since performing online CL experiments on large-scale datasets is computationally expensive, we choose a small dataset, **CUB-200** as the 2nd dataset after ImageNet-1K. CUB-200 is more challenging than other small datasets (Lee et al., 2023). After being pre-trained on ImageNet-1K, a model learns CUB-200 in *sample-by-sample* manner, i.e., after receiving a new sample, it takes one SGD step using 51 samples (50 old samples and 1 new sample). Storage is constrained by a maximum number of samples, i.e., 80K and 20K, corresponding to 6.2% and 1.5% of the entire dataset (ImageNet and CUB combined), respectively. As shown in Table 6, SGM combined with REMIND outperforms standalone REMIND by large margins in all metrics. When storage is bounded by 20K samples, SGM improves REMIND's final accuracy by absolute 8.64% and reduces REMIND's stability gap by a factor of 3.95×.

## 6.4 Supporting Studies

We include additional supporting studies in the Appendix and summarize the findings here.

**Non-Rehearsal Methods.** While non-rehearsal methods are less performant for CL, it is interesting to study SGM when combined with non-rehearsal methods to determine if our findings are consistent. During CIL of ImageNet-1K and Places365-LT, SGM also benefits a non-rehearsal method, LwF (Li & Hoiem, 2017) with an absolute gain of 35.24% in final accuracy and a 2.6× reduction in the stability gap (see Appendix E).

**Class-Balanced Rehearsal.** Since our main experiments are based on conventional rehearsal without class balance, we also conduct experiments using class-balanced rehearsal. Compared to vanilla rehearsal, SGM reduces the stability gap by 7.3× when both use class-balanced rehearsal. SGM also achieves continual knowledge transfer (see Appendix D.2).

**Supervised Pre-Training.** In our main results, SGM reduces the stability gap using the self-supervised pre-trained ConvNeXtV2 model, and self-supervised models are known to perform better in CL (Gallardo et al., 2021). We also conduct experiments with the supervised pre-trained ConvNeXtV1 (Liu et al., 2022) to examine how SGM performs without a self-supervised backbone. We find that SGM is effective without the self-supervised backbone. Compared to vanilla rehearsal, SGM achieves 6× more stability, 3.9× more plasticity, and 35× more continual knowledge transfer when both use ConvNeXtV1 (see Appendix D.3).

**Vision Transformers.** We also study the behavior of SGM with ViTs, which now rival CNNs. Compared to vanilla rehearsal, SGM reduces the stability gap by 2.4× when both use a pre-trained ViT (see Appendix D.4).

# 7 Limitations & Future Work

Our work is built upon the assumption that a well-trained DNN is provided and performance on the original tasks must be maintained or improved. If this assumption is violated, as seen in some CL papers that do not start with pre-trained models, SGM's dependence on LoRA would impair its utility. Although most of the components of SGM, such as weight initialization, dynamic soft targets, and OOCF, can be readily applied to

DNNs trained from scratch, LoRA cannot be used because DNN layers without LoRA would remain frozen and unlearned. Applying SGM in this setting would require a variant of LoRA where LoRA could be used for all layers from the start of training such that the DNN rivaled a jointly trained model without LoRA. We used pre-trained ConvNeXtV1, ConvNeXtV2, and MobileViT DNNs, which are amenable to LoRA. Future work could investigate different LoRA variants and network adapters (Han et al., 2024) with SGM. It would also be interesting to study the efficacy of SGM for meeting the needs of CL for embedded devices Hayes & Kanan (2022).

Our study focused on image classification. Future work could explore SGM's efficacy for regression, object detection (Acharya et al., 2020; 2019), reasoning Hayes & Kanan (2021), and semantic segmentation (Zhang et al., 2022). CL methods developed only on small datasets often do not scale to larger datasets such as ImageNet-1K (Zhou et al., 2023), so we studied a combination of ImageNet-1K and Places365. Due to computational limitations and a lack of readily available large-scale image classification datasets, we could not study CL for more than 1365 classes. In future work, assessing how well SGM scales with increasing dataset sizes and class counts would be valuable. We hope our work will inspire future research to find better approaches to prevent model decay due to concept drift.

## 8 Conclusion

Although there has been remarkable progress in mitigating catastrophic forgetting, almost all CL methods remain unsuited for practical applications such as addressing model decay (Verwimp et al., 2024; Harun et al., 2023a). To eliminate the need for frequent retraining of deployed DNNs, CL has to combat model decay, mitigate the stability gap, and be more computationally efficient than retraining from scratch in large-scale CL settings. Focusing on this scenario, we showed that SGM meets these criteria. With the growing energy usage of deep learning models (Luccioni et al., 2022; Patterson et al., 2021; Wu et al., 2022), reducing the need for retraining from scratch could significantly contribute to reducing carbon emissions associated with training DNNs. Our work aligns CL with these real-world challenges, and we hope it encourages the CL community to focus on them.

**Acknowledgments**

This work was supported in part by NSF awards #1909696, #2326491, #2125362, and #2317706. The views and conclusions contained herein are those of the authors and should not be interpreted as representing any sponsor's official policies or endorsements. We thank Jhair Gallardo, Junyu Chen, Shikhar Srivastava, and Robik Shrestha for their feedback and comments on the manuscript.

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

# Appendix

## A  Implementation Details and Additional Results

We organize additional supporting experimental findings as follows:

- Appendix B provides details on the datasets used in this paper.

- Appendix C provides additional implementation and training details for all of the methods.

- Appendix D provides additional CIL experiments and results with rehearsal, including an analysis of learning curves, studying alternative sampling strategies for rehearsal, using a non-self-supervised backbone CNN, using a vision transformer, and using a balanced dataset. We find that SGM works well across these experiments and analyses compared to vanilla rehearsal.

- Appendix E studies the behavior of our stability gap mitigation method when used with Learning without Forgetting (LwF), a popular regularization method used in CL. We find that our method greatly improves results, illustrating that the mitigation strategy is not specific to rehearsal.

- Appendix F demonstrates that our evaluation metrics facilitate a more accurate and meaningful comparison between different CL models.

- Appendix G illustrates the process of updating soft targets.

- Appendix H includes additional discussion.

## B  Dataset Details

In this paper, we use two large-scale datasets (**ImageNet-1K** and **Places365-Standard**) and two small-scale datasets (**Places365-LT** and **CUB-200**). ImageNet-1K (Russakovsky et al., 2015) has 1.28 million images from 1000 categories, each with $732 - 1300$ training images and 50 validation images. Places365-LT (Liu et al., 2019) is a long-tailed dataset with an imbalanced class distribution. It is a long-tailed variant of the Places-2 dataset (Zhou et al., 2017). Places365-LT has 365 classes and 62500 training images with 5 to 4980 images per class. For its test set, we use the Places365-LT validation set from (Liu et al., 2019) which consists of a total of 7300 images with a balanced distribution of 20 images per class. Places365-Standard (Zhou et al., 2017) has over 1.8 million training images from 365 classes with $3068 - 5000$ images per class. We use the validation set consisting of 100 images per class to test the models. CUB-200 (Wah et al., 2011) has RGB images of 200 bird species with 5994 training images and 5794 test images.

## C  Additional Implementation Details

In this section, we provide additional implementation details for the models.

**Main Experiments.** For both CIL and IID experiments, we train SGM with rehearsal, vanilla rehearsal, and output layer only using cross-entropy loss for 600 iterations per rehearsal session. During each iteration model is updated on 128 samples. All methods use the same ConvNeXtV2 backbone [3], use AdamW optimizer with weight decay of 0.05 and initial learning rates of $10^{-3}$ (SGM and vanilla) and $10^{-2}$ (output layer only). The learning rate is reduced in earlier layers by a layer-wise decay factor of 0.9. The learning rate scheduler is not applied for vanilla and output layer only due to poor performance. On the other hand, SGM uses OneCycle learning rate scheduler (Smith & Topin, 2017). The joint model (upper bound) is trained for 12500 iterations on all data i.e., ImageNet-1K and Places365-LT combined using an initial learning rate of $10^{-4}$ without a scheduler. For all experiments, we set the rank of the LoRA weight matrices to 48. In all cases, all metrics are based on Top-1 accuracy (%). Most CL experiments including those in Sec. 5, and Sec. 6.1 adhere to the aforementioned settings unless otherwise noted.

---

[3]Pre-trained weights are available here: `https://github.com/facebookresearch/ConvNeXt-V2`

**Storage Constrained Offline CL with DERpp and GDumb.** We describe settings used in Sec. 6.2 where we combine SGM with DERpp and GDumb while using identical settings e.g., the same ImageNet-1K pre-trained ConvNeXt V2 Femto network and the same optimizer settings. Each model pre-trained on ImageNet-1K learns Places365-Standard in 5 batches subsequently (73 categories per batch). Each rehearsal session performs a total of 1200 iterations with 256 samples per iteration. DERpp employs distillation and regularization along with rehearsal to prevent catastrophic forgetting. It regularizes loss on old samples and uses an additional distillation loss on logits of old samples for promoting consistency. We set coefficients $\alpha = 0.1$ and $\beta = 0.9$ for distillation and regularization respectively. GDumb randomly removes a sample from the largest class when the buffer reaches its maximum capacity and maintains a class-balanced buffer. For all methods, the buffer is bounded by a maximum number of samples (80% ImageNet-1K + 20% Places365-Standard). DERpp, GDumb, and SGM use an initial learning rate of $1 \times 10^{-3}$, $1 \times 10^{-3}$, and $1.5 \times 10^{-3}$, respectively, for batch size 256. The joint model (upper bound) uses an initial learning rate of $10^{-2}$ and 12K iterations with 256 samples per iteration. We assess performance during rehearsal every 100 mini-batches to compute the metrics.

**Class-balanced Rehearsal.** For class-balanced rehearsal experiments in Appendix D.2, SGM with rehearsal and vanilla rehearsal use an initial learning rate of $10^{-3}$ and $10^{-4}$, respectively.

**Non-SSL Backbone CNN.** For non-SSL backbone experiments with ConvNeXt V1-Tiny (Liu et al., 2022) in Appendix D.3, initial learning rates for SGM with rehearsal, vanilla rehearsal, and joint model are $4 \times 10^{-3}$, $3 \times 10^{-3}$, and $10^{-4}$ respectively. ConvNeXt V1-Tiny has been pre-trained on ImageNet-1K using supervised learning [4].

**ViT Backbone.** For ViT backbone experiments in Appendix D.4, we select MobileViT-Small (Mehta & Rastegari) (5.6M) pre-trained on ImageNet-1K using supervised learning [5]. For universal feature extraction, we freeze the first 8 blocks including stem, 6 MobielNetV2 blocks, and 1 MobileViT block. We keep the remaining blocks (1 MobileNetV2 block and 2 MobileViT blocks) and layers (1 CNN layer and 1 linear layer) plastic which correspond to 96.4% of the total parameters. We apply LoRA (rank=48) to query, key, and value projection matrices in the self-attention module of MobileViT transformer blocks. All methods use the AdamW optimizer with a weight decay of 0.01. Vanilla rehearsal and SGM with rehearsal use an initial learning rate of $3 \times 10^{-3}$ and $4 \times 10^{-3}$, respectively. The initial learning rate for the joint model is $10^{-2}$. Places365-LT data is learned over 5 rehearsal sessions (73 classes per session) where each session includes 1200 iterations with 32 samples per iteration. The joint model is trained for 25K iterations with 64 samples per iteration. All other settings are identical to those in the main experiments.

**Balanced (Non-LT) Dataset.** For experiments with a balanced dataset in Appendix D.5, SGM with rehearsal and vanilla rehearsal use an initial learning rate of $1.5 \times 10^{-3}$ and $10^{-3}$, respectively. Each model is pre-trained on ImageNet-1K and then learns Places365-Standard in 5 batches subsequently (73 categories per batch). Each rehearsal session has a total of 1200 iterations with 256 samples per iteration. We set $f = 0.5$ for compute budget. Hence, at the end of CL, the total number of SGD model updates is 50% of the total number of samples in the entire dataset (ImageNet and Places-Standard combined). We assess performance during rehearsal every 100 minibatches to compute the metrics. The joint model uses an initial learning rate of $10^{-2}$ and 12K iterations with 256 samples per iteration.

**Storage Constrained Online CL with REMIND.** In Appendix 6.3, we use identical settings and hyperparameters for both REMIND and REMIND + SGM methods. We use ImageNet-1K pre-trained ConvNeXt V2 Femto with similar network configurations and LoRA configurations as used in main experiments. We use the AdamW optimizer and REMIND's default per-class learning rate scheduler. We set the initial learning rate to $1 \times 10^{-3}$, the final learning rate to $1 \times 10^{-5}$, and weight decay to 0.05. Following REMIND, we perform rehearsal with a mini-batch of 51 samples including 50 old samples and 1 new sample. Each method learns the CUB-200 dataset in *sample-by-sample* manner. For all methods, the buffer is bounded by a maximum number of samples (75% − 93% ImageNet).

**Non-rehearsal Methods.** In Appendix E, LwF has similar configurations as vanilla rehearsal except for the initial learning rate ($6 \times 10^{-5}$). LwF + SGM has a similar configuration as SGM with rehearsal except

---

[4] The pre-trained weights are available here: https://github.com/facebookresearch/ConvNeXt
[5] The pre-trained weights are available here: https://github.com/apple/ml-cvnets

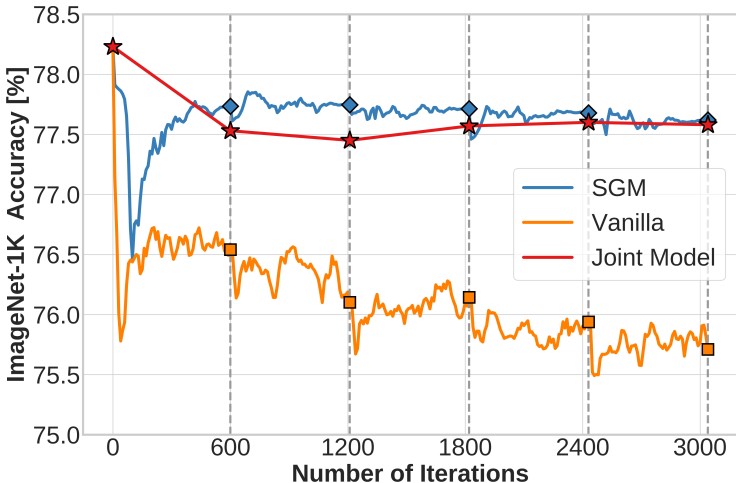

Figure 4: **Stability gap over all rehearsal sessions.** After pre-training on ImageNet-1K, the model learns 365 new classes from Places365-LT over five rehearsal sessions (600 iterations per rehearsal session). SGM quickly recovers old performance at the beginning of CL whereas vanilla fails to obtain full recovery. After each rehearsal session (vertical dotted gray line), the final top-1 accuracy (%) is highlighted by diamond (SGM), star (joint model), and square (vanilla). The joint model (upper bound) is jointly trained on ImageNet and seen CL batches from the Places dataset.

the initial learning rate ($2 \times 10^{-4}$). During each iteration model is updated on 64 new samples without any rehearsal of old samples. Our mitigation approaches e.g., dynamic soft targets, data-driven weight initialization, OOCF, and LoRA are applied for LwF similarly as they are applied for rehearsal methods in main experiments.

**Weight Initialization.** For SGM, we use our data-driven weight initialization (Sec. 4) to initialize the weights in the final output layer. For other compared methods, we use He initialization (He et al., 2015) to initialize the weights in the final output layer.

All other settings adhere to the above-mentioned general settings for the main experiments unless otherwise mentioned. Hyperparameters are tuned to maximize performance for each method.

**Compute.** We ran all experiments on the same hardware with a single GPU (NVIDIA RTX A5000).

# D   Additional CIL Analysis & Experiments

In this section, we conduct additional analysis of the CIL experiments in the main results as well as present additional experiments. Implementation details and dataset details are given in Appendix C and B.

## D.1   Qualitative Analysis

In this section, we present a qualitative analysis to see how SGM mitigates the stability gap and enhances computational efficiency.

### D.1.1   Stability Gap Over All Rehearsal Sessions

In our main text, our figures are averaged across rehearsal sessions. In Fig. 4, we instead present all of the learning curves in sequence, where we denote when the next batch containing new classes is received.

When rehearsal begins, accuracy on ImageNet-1K for vanilla rehearsal drops dramatically and gradually decreases throughout the rehearsal sessions. In the end, vanilla fails to recover the original performance using a total of 3000 iterations. In contrast, SGM shows better performance throughout rehearsal sessions

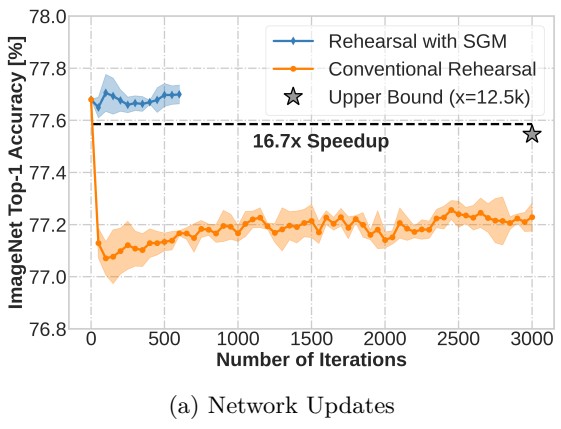 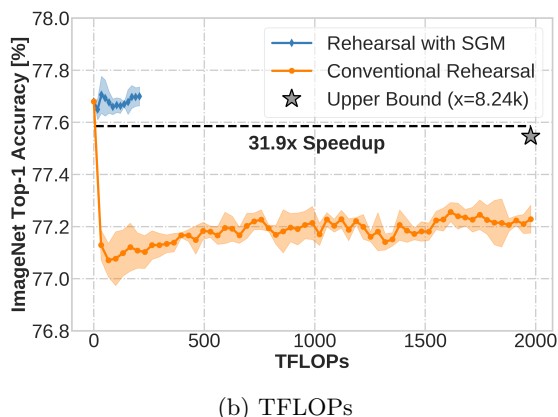

(a) Network Updates

(b) TFLOPs

Figure 5: **Computational efficiency.** Our method, SGM, provides a 16.7× speedup in the number of network updates and a 31.9× speedup in TFLOPs compared to a joint model (upper bound) with the combined 1365 class dataset (ImageNet-1K and Places365-LT combined). For SGM and conventional rehearsal, we show the stability gap in the learning curve averaged over rehearsal sessions.

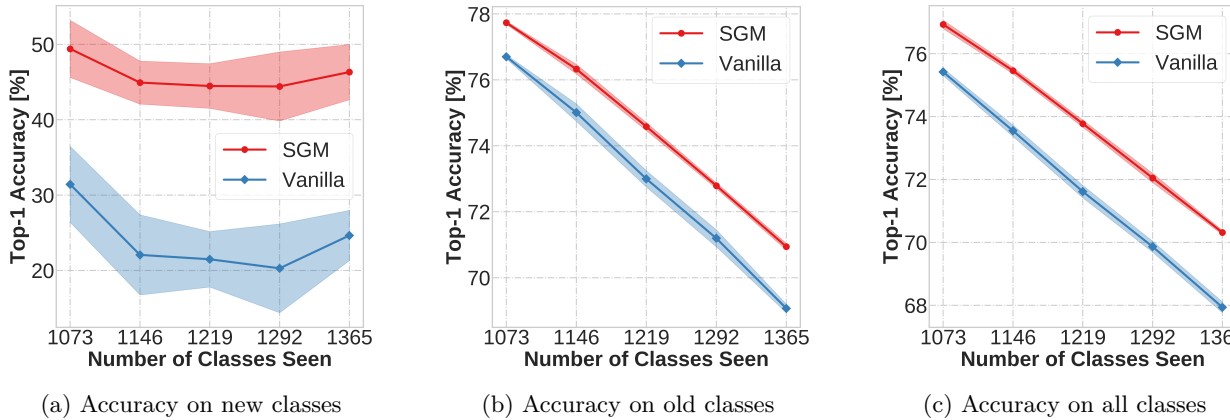

(a) Accuracy on new classes

(b) Accuracy on old classes

(c) Accuracy on all classes

Figure 6: **Stability-plasticity.** After pre-training on ImageNet-1K, the model learns 365 new classes from Places365-LT over five batches (73 new classes per batch) in CIL setting. The accuracy is averaged over 6 runs and shaded region indicates standard deviation.

with reduced stability gap and full recovery compared to the joint model. Models are evaluated every 10 iterations. After each rehearsal session, SGM outperforms vanilla and matches or exceeds the accuracy of the joint model (upper bound).

### D.1.2   Computational Efficiency

To measure computational efficiency, we consider two criteria: 1) number of network updates (Sec. 3.1) and 2) FLOPs (floating-point operations). For FLOPs analysis, we use DeepSpeed [6] with the same GPU across compared models. We perform computational analysis for CIL on a combination of ImageNet-1K and Places365-LT datasets.

As shown in Fig. 5, SGM provides a 16.7× speedup in network updates and a 31.9× speedup in TFLOPs compared to a joint model (upper bound). SGM fully recovers the original performance on ImageNet-1K using 600 iterations per rehearsal session whereas conventional rehearsal fails to do that using 3000 iterations per rehearsal session.

---

[6]https://github.com/microsoft/DeepSpeed

Table 7: **Class Balanced Rehearsal**. A model pre-trained on ImageNet-1K learns Places365-LT in 5 rehearsal sessions. Here $\mu$ denotes average accuracy (%) over rehearsal sessions and $\alpha$ is final accuracy (%) on 1365 classes. $\#P$ denotes total number of trainable parameters in Millions.

| Method | $\#P\downarrow$ | $\mathcal{S}_\Delta\downarrow$ | $\mathcal{P}_\Delta\downarrow$ | $\mathcal{CK}_\Delta\downarrow$ | $\mu\uparrow$ | $\alpha\uparrow$ |
|---|---|---|---|---|---|---|
| Joint Model (Upper Bound) | 5.08 | — | — | — | — | 70.69 |
| Vanilla Rehearsal | 5.08 | 0.022 | 0.316 | 0.021 | 72.24 | 69.03 |
| **SGM** (Ours) | **1.45** | **0.003** | **0.089** | **−0.003** | **74.00** | **70.61** |

Table 8: **CIL without Self-Supervised Pre-Training**. This table shows results from ConvNeXt V1-Tiny pre-trained on ImageNet-1K using supervised learning, which then learns Places365-LT in 5 rehearsal sessions in CIL setting. Here $\mu$ denotes average accuracy (%) over rehearsal sessions and $\alpha$ is final accuracy (%) on 1365 classes. $\#P$ denotes total trainable parameters in Millions.

| Method | $\#P\downarrow$ | $\mathcal{S}_\Delta\downarrow$ | $\mathcal{P}_\Delta\downarrow$ | $\mathcal{CK}_\Delta\downarrow$ | $\mu\uparrow$ | $\alpha\uparrow$ |
|---|---|---|---|---|---|---|
| Joint Model (Upper Bound) | 27.00 | — | — | — | — | 74.16 |
| Vanilla Rehearsal | 27.00 | 0.030 | 0.396 | 0.035 | 74.73 | 70.67 |
| **SGM** (Ours) | **3.53** | **0.005** | **0.102** | **0.001** | **77.48** | **73.92** |

### D.1.3 Stability-Plasticity

In Fig. 6, we also illustrate the model's accuracy on new, old, and all classes in all rehearsal sessions where SGM achieves higher accuracy than vanilla. This indicates that SGM consistently improves the model's plasticity (Fig. 6a), stability (Fig. 6b), and knowledge accumulation (Fig. 6c).

### D.2 Class Balanced Uniform Sampling for Rehearsal

In our main results, we sampled randomly during rehearsal without balancing for each class. However, prior work has shown that class-balanced random sampling works significantly better than unbalanced uniform sampling for long-tailed datasets (Harun et al., 2023b). We conducted CIL experiments to examine this in our unlimited storage for rehearsal setup where we learned ImageNet-1K followed by Places365-LT.

Table 7 shows that using class-balanced rehearsal, SGM improves performance in most criteria compared to previous results without class balance (Table 1). When both vanilla and SGM use class balanced rehearsal, SGM outperforms vanilla by $7.3\times$ in stability gap, $3.6\times$ in plasticity gap and provides continual knowledge transfer ($\mathcal{CK}_\Delta < 0$).

### D.3 Analysis with a Non-Self-Supervised Backbone CNN

Much of deep learning has moved toward self-supervised pre-training prior to supervised fine-tuning, especially in foundation models (Devlin et al., 2018; Brown et al., 2020; Ramesh et al., 2021), since this has been shown to reduce overfitting on the pretext dataset used for self-supervised learning and to generalize better to downstream tasks. In the main text, we used the self-supervised ConvNeXtV2 architecture. This may have enabled our system to achieve higher results on Places365-LT than if the CNN was initialized from ImageNet-1K with supervised learning. To determine if our general trends for the methods hold, we conducted another experiment with ConvNeXtV1-Tiny (29M), which is pre-trained on ImageNet-1K without self-supervision. We conducted CIL experiments to examine ConvNeXtV1 models in our unlimited storage for rehearsal setup where we learned ImageNet-1K followed by Places365-LT.

Experimental results in Table 8 demonstrate that SGM with a supervised backbone mitigates the stability gap and enhances performance in all criteria. Therefore the efficacy of SGM does not depend upon self-supervised pre-training.

### D.4 Analysis with using a Vision Transformer Backbone

In this section, we study the behavior of the system for a ViT model pre-trained with supervised learning. For this, we select a lightweight transformer, MobileViT-Small (Mehta & Rastegari). MobileViT learns local and global representations using convolutions and transformers, respectively. It has a total of 5.6 million parameters and top-1 accuracy of 78.4% on ImageNet-1K. We conducted CIL experiments to examine MobileViT models in our unlimited storage for rehearsal setup where we learned ImageNet-1K followed by Places365-LT.

Table 9 shows the comparison between vanilla and SGM when they have the same MobileViT backbone. SGM shows better performance in all criteria using $3.8\times$ fewer parameters than vanilla rehearsal.

Table 9: **Vision Transformer Backbone**. A model pre-trained on ImageNet-1K learns Places365-LT in 5 rehearsal sessions. Here $\mu$ denotes average accuracy (%) over rehearsal sessions and $\alpha$ is final accuracy (%) on 1365 classes. $\#P$ denotes the total number of trainable parameters in Millions.

| Method | $\#P\downarrow$ | $\mathcal{S}_\Delta\downarrow$ | $\mathcal{P}_\Delta\downarrow$ | $\mathcal{CK}_\Delta\downarrow$ | $\mu\uparrow$ | $\alpha\uparrow$ |
|---|---|---|---|---|---|---|
| Joint Model (Upper Bound) | 4.97 | — | — | — | — | 69.10 |
| Vanilla Rehearsal | 4.97 | 0.039 | 0.434 | 0.046 | 70.18 | 66.35 |
| **SGM** (Ours) | **1.30** | **0.016** | **0.140** | **0.016** | **72.09** | **67.96** |

### D.5 Balanced (Non-LT) Dataset

In a real-world setting, data distribution is commonly long-tailed and imbalanced, hence we used Places365-LT dataset in the main results. However, our analysis holds for balanced and non-LT datasets as well. We study this using Places365-Standard. We omit storage constraints in this experiment to solely focus on mitigation methods without the influence of other variables. Results in Table 10 show that SGM outperforms vanilla rehearsal in all criteria.

Table 10: **Non-LT Dataset**. A model pre-trained on ImageNet-1K learns Places365-Standard in 5 rehearsal sessions in CIL setting. Here $\mu$ denotes average accuracy (%) over rehearsal sessions and $\alpha$ is final accuracy (%) on 1365 classes. $\#P$ denotes the total number of trainable parameters in Millions.

| Method | $\#P\downarrow$ | $\mathcal{S}_\Delta\downarrow$ | $\mathcal{P}_\Delta\downarrow$ | $\mathcal{CK}_\Delta\downarrow$ | $\mu\uparrow$ | $\alpha\uparrow$ |
|---|---|---|---|---|---|---|
| Joint Model (Upper Bound) | 5.08 | — | — | — | — | 65.37 |
| Vanilla Rehearsal | 5.08 | 0.078 | 0.201 | 0.082 | 66.37 | 56.63 |
| **SGM** (Ours) | **1.45** | **0.054** | **0.091** | **0.047** | **68.47** | **59.21** |

## E SGM Enhances Non-rehearsal Methods

We hypothesized that SGM would be helpful for non-rehearsal methods as well. We, therefore, study SGM using Learning without Forgetting (LwF) (Li & Hoiem, 2017), which pioneered using knowledge distillation in CL (Zhou et al., 2023). Instead of rehearsal, LwF stores a copy of the model before learning the new CL batch to update the model with distillation. LwF has been shown to reduce catastrophic forgetting in a range of CL scenarios, although it and other regularization-based methods have not been shown to be effective in the CIL setting (Zhou et al., 2023).

We conducted an experiment to compare vanilla LwF with a version of LwF that uses SGM without rehearsal during CIL of ImageNet-1K and Places365-LT. For this, we use **ConvNeXtV2-Femto** pre-trained on ImageNet-1K using SSL. After being pre-trained on ImageNet-1K, the model sequentially learns 5 incremental batches of data (73 classes per batch) from Places365-LT in offline CL with CIL ordering. The training is compute-constrained where the model is updated over 600 minibatches. Each minibatch consists of 64 new samples and rehearsal of old samples is omitted. Additional implementation details are given in Appendix C.

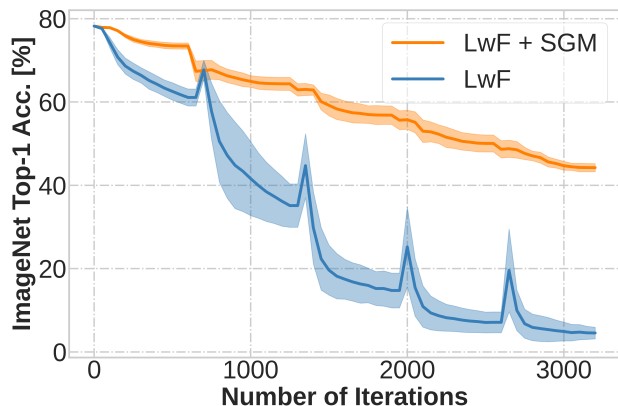

Figure 7: **Comparison with non-rehearsal method.** The Y-axis shows an average accuracy (%) of 6 runs with a standard deviation (shaded region). The network is trained on ImageNet-1K and then learns 365 new classes from Places-LT over five batches (73 new classes and 600 iterations per batch). When a new batch arrives, accuracy on ImageNet-1K for LwF plummets. LwF fails to recover performance and ends up with a large stability gap. In contrast, LwF with SGM does not plummet like LwF and shows better performance throughout the CL phase with a significantly reduced stability gap.

Overall results are given in Table 11 and a learning curve is given in Fig. 7. As expected based on prior results, rehearsal methods vastly outperform LwF; however, we find that SGM provides an enormous benefit to LwF in terms of reducing the stability gap, resulting in increased accuracy.

Table 11: **Comparison with non-rehearsal method**. A model pre-trained on ImageNet-1K learns Places365-LT in 5 rehearsal sessions in CIL setting. Results are averaged over 6 runs. Here $\mu$ denotes average accuracy (%) over rehearsal sessions and $\alpha$ is final accuracy (%) on 1365 classes. $\#P$ denotes total trainable parameters in Millions. For the non-rehearsal baseline, we select LwF that regularizes the model based on knowledge distillation.

| Method | $\#P \downarrow$ | $\mathcal{S}_\Delta \downarrow$ | $\mathcal{P}_\Delta \downarrow$ | $\mathcal{CK}_\Delta \downarrow$ | $\mu \uparrow$ | $\alpha \uparrow$ |
|---|---|---|---|---|---|---|
| Joint Model (Upper Bound) | 5.08 | — | — | — | — | 70.69 |
| Vanilla Rehearsal | 5.08 | 0.020 | 0.385 | 0.031 | 71.68 | 67.94 |
| **SGM** (Ours) | **1.45** | **0.001** | **0.087** | **0.002** | **73.71** | **70.31** |
| LwF | 5.08 | 0.605 | 0.450 | 0.607 | 24.04 | 4.76 |
| **LwF + SGM** (Ours) | **1.45** | **0.236** | **0.072** | **0.235** | **54.87** | **40.00** |

## F  Reasons for New Metrics

In the main text in Sec. 3.2, we proposed new metrics that are normalized against a joint model (upper bound). Unlike earlier metrics (De Lange et al., 2023), our metrics enable a more significant and accurate comparison between *different* CL models. Moreover, our metrics can be used to analyze whether the CL model is meeting the needs of the industry to catch up to the joint model retrained from scratch when new data becomes available.

In this section, we illustrate two cases using synthetic examples in Fig. 8 to build intuitions. In each plot, we show the drop in old task accuracy when updating models on a new task. We compare two CL models in terms of the stability gap i.e., drop in old task accuracy, to find a better model.

**Case 1.** The metric proposed in earlier work (De Lange et al., 2023) measures the stability gap of a model by evaluating the largest drop in old task performance compared to the *same* model's performance. According to their metric, CL model 1 is considered to be better than CL model 2 as it exhibits a smaller drop in accuracy for the old task, as shown in Fig. 8(a). However, in contrast to this observation, it can be observed

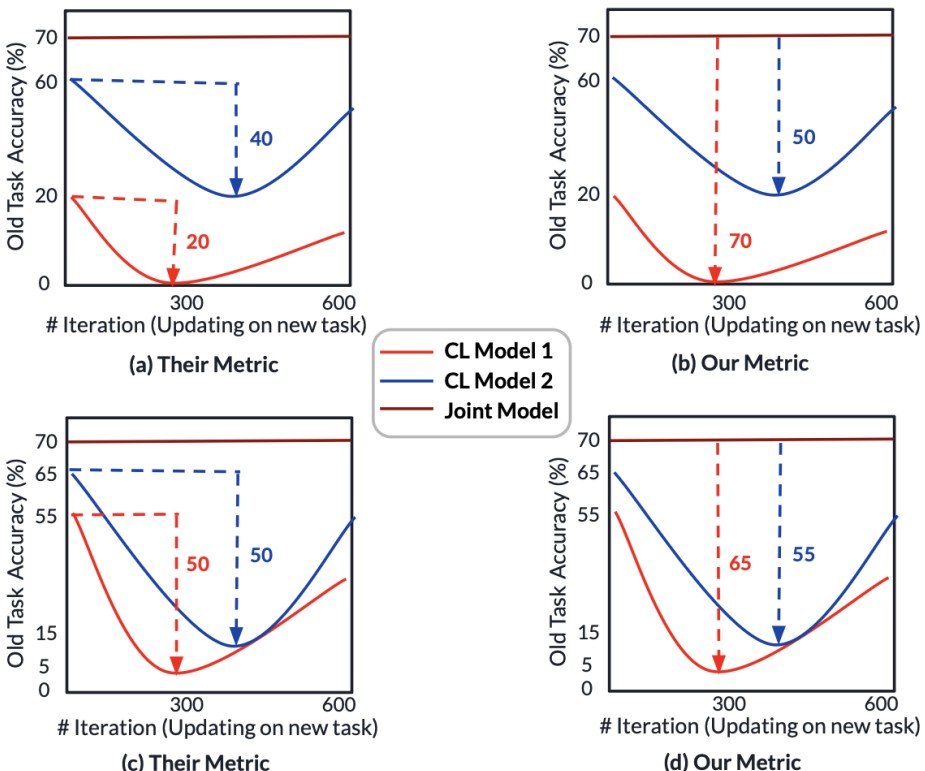

Figure 8: **Efficacy of our metric.** We present a series of synthetic plots that display the performance of a model on an old task as it learns a new task. The largest drop in accuracy on the old task is marked by a dotted line with an arrow. This drop in accuracy is also displayed as a number for ease of comparison. In sub-figures (a) and (b), our metric delivers a different outcome from theirs (prior work). In sub-figures (c) and (d), unlike their metric, our metric can distinguish between two CL models.

that CL model 2 performs better than CL model 1 as it shows a smaller drop in old task accuracy compared to CL model 1 while employing the joint model (upper bound) as a universal reference point. This is captured by our metric and can be seen in Fig. 8(b).

**Case 2.** We demonstrate that the metric proposed in previous work (De Lange et al., 2023) is unable to differentiate between two CL models that show the same decline in old task performance, as depicted in Fig. 8(c). In contrast, our metric, as illustrated in Fig. 8(d), is capable of identifying that CL model 2 (with a smaller drop) performs better than CL model 1 (with a larger drop). From this, it is evident that different CL models cannot be compared without a universal upper bound.

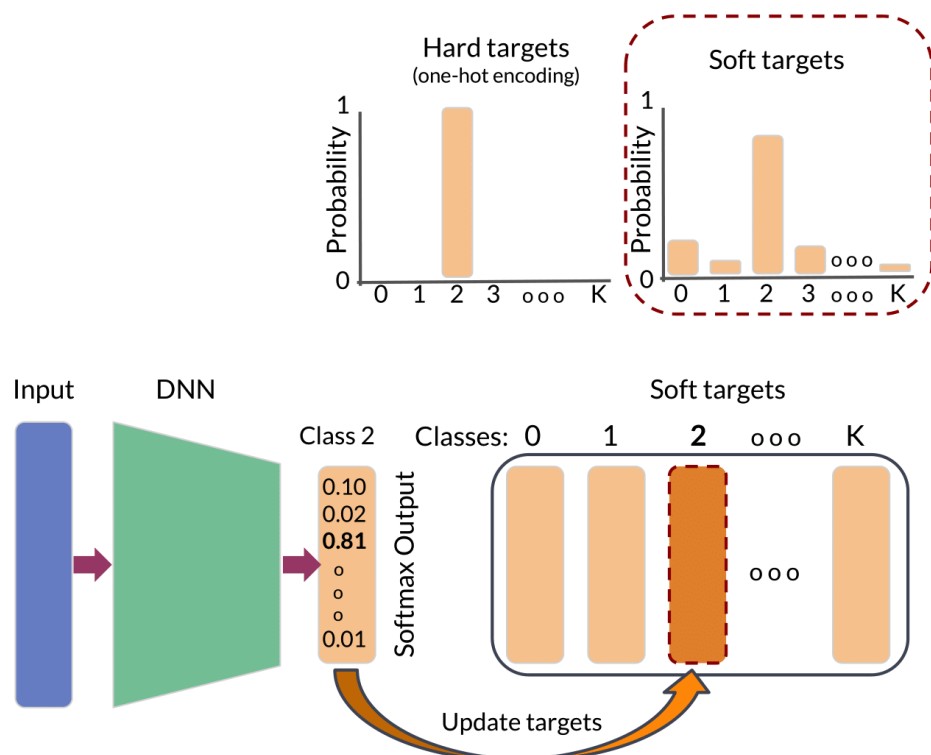

Figure 9: **Process of updating soft targets.** Here we illustrate the update mechanism of our dynamic soft targets. For each class $k$, we maintain a probability distribution over total $K$ classes, denoted by $\mathbf{u}_k \in \mathbb{R}^K$. During iteration $i$, we update this vector $\mathbf{u}_k^i$ if the model ($\theta_i$) correctly predicts the class $k$. In the following iteration $i+1$, $\mathbf{u}_k^i$ will serve as soft targets to train the model, $\theta_{i+1}$. Unlike hard targets, soft targets do not enforce strict inter-class independence and prevent the stability gap.

## G  Soft Targets Update

In the main text in Sec. 4, we proposed dynamic soft targets to prevent the stability gap. In this section, we illustrate how we update the soft targets in Fig. 9. Soft targets are updated using the model's predicted probabilities.

## H  Additional Discussion

**What is the connection between evaluation metrics and stability gap mitigation approaches?**

In the main text in Sec. 3.2, we formalized our evaluation metrics to measure the stability gap ($\mathcal{S}_\Delta$), plasticity gap ($\mathcal{P}_\Delta$), and continual knowledge gap ($\mathcal{CK}_\Delta$). We found that our proposed mitigation methods in Sec. 4 effectively mitigated the stability, plasticity, and continual knowledge gaps when evaluated using metrics from Sec. 3.2. Therefore, the evaluation metrics can be regarded as the tools to measure the efficacy of the proposed mitigation methods.

Mitigation methods were designed to prevent *factors* contributing to the stability gap such as large loss at the output layer and excessive network plasticity. All three metrics were used in all evaluations and experiments in the main paper and the Appendix.

