# OpenReview forum: "Overcoming the Stability Gap in Continual Learning"
_TMLR — Accepted by TMLR_

### Review · Reviewer_5mcy · 2024-06-07

**Summary Of Contributions:**

This work focuses on continual learning, where the task changes while a model is already deployed, and the model needs to learn the new task while not losing overall performance. The work proposes several components to mitigate the performance drop: dynamic soft targets and weight initialization to attempt at reducing the loss of the model, and OOCF and LoRA to attempt at reducing the plasticity of the model. All the components are shown to effectively reduce the performance drop and computational overhead, and are combined into the SGM method that further achieves these benefits.

**Audience:**

Yes

**Broader Impact Concerns:**

I do not have broader impact concerns to attribute to this work.

**Claims And Evidence:**

No

**Requested Changes:**

I do not see how the results validate the hypothesis, only that the proposed method (which is indeed inspired by the hypothesis) performs well, so I think there needs to be a clarification to this point, or the text altered. It is very important for me that this point is addressed.

Minor:
- Why are the values on the x-axis of Figures 1a and 1b different?

**Strengths And Weaknesses:**

The problem the work focuses on in relevant for both industry and academia, paper is well and clearly written, and the experiments are satisfactory in showing the benefits of the proposed approaches.

Each component of the method does not seem to be particularly novel, especially the most impactful LoRA, but the overall method is well motivated, validated and explained. I did not understand what is meant by "hypothesis-driven" experiments. Specifically, the paper writes two hypothesis, which are that high losses *cause* the stability gap and that high plasticity *causes* the stability gap. Then, the paper proposes additions to the baseline *inspired* by these hypothesis, and show that they improve over the baseline. However, I do not think this shows that the hypothesis were confirmed, because plasticity and loss can be *correlated* with the stability gap, not causing it. So I think it is very important that the authors either explain what they mean by "hypothesis-driven" experiments or remove the nomenclature.

Due to my last concern, I am claiming "no" to the "claims and evidence" question, even though I am open for clarification from the authors, or a change in the paper.

---

> ### Author Response · Authors · 2024-06-07
> **Author Response**
>
> Thank you for your insightful comments and feedback.
> We have carefully considered your concerns and tried to address them.
> Below, we have provided detailed responses to each review separately.
>
> **W1. Is there novelty in the proposed methods?**
>
> - Although LoRA was proposed by others and is widely used in the NLP domain and transformer-based models, to the best of our knowledge, we are the first to incorporate LoRA into CNNs. Each component of SGM has novelty in the context of the stability gap. By combining these four components, SGM stands out as a unique and innovative method. However, we did not explicitly mention SGM as a _novel_ method in the paper.
> - Additionally, as shown in Sec. 3.2, we proposed three novel metrics to measure the stability gap, plasticity gap, and continual knowledge gap.
> - Our findings are also novel, and we hope our work will inspire future research to address model decay due to concept shift.
>
>
> **W2. Did the results validate the hypotheses?**
>
> - Our study utilized two hypotheses to understand the causes of the stability gap in our setting.
> Our results supported these hypotheses. As shown in Fig. 2(a) and 2(b), a large initial loss due to learning a new task exacerbated the stability gap. However, our proposed methods—dynamic soft targets and data-driven weight initialization—reduced this loss and, consequently, the stability gap.
>
> - On the other hand, as shown in Fig. 2(c), increasing network plasticity exacerbated the stability gap, while constraining plasticity using LoRA mitigated it. These findings support our hypothesis that large loss and increased plasticity contribute to the stability gap.
>
> - Regarding your concerns about whether loss and plasticity __cause__ the stability gap, we have revised the relevant parts in the abstract, introduction, and other sections to indicate that loss and plasticity __increase__ the stability gap. We have highlighted the modified text in blue. Please let us know if this addresses your concerns. We appreciate your feedback and the opportunity to provide clarifications.
>
> **W3. Why did the authors mention "hypothesis-driven" experiments?**
>
> - We referred to the aforementioned experiments as "hypothesis-driven" experiments because they aimed to elucidate factors contributing to the stability gap. There was a mention of hypothesis-driven experiments in Fig.1 caption.
> Following your suggestion, we have now eliminated the nomenclature and adjusted the text.
>
> **W4. How is the proposed method related to the hypotheses?**
>
> - As described in Section 4, we proposed four mitigation mechanisms: two (data-driven weight initialization and dynamic soft targets) to reduce loss due to new tasks and two (LoRA and OOCF) to restrict network plasticity, aligning with our hypotheses. Incorporating these mechanisms, our method, SGM, consistently reduced the stability gap across various settings and improved performance in all criteria.
>
> **Requested Changes**
>
> - **How did the results validate the hypothesis?** We have tried to explain this in our responses above and modified text in the paper.
>
> - **Why are the values on the x-axis of Figures 1a and 1b different?** In Fig. 1(a), we show the stability gap in the learning curve across all five rehearsal sessions, each comprising 600 training iterations for a total of 3000 iterations. In Fig. 1(b), we show the stability gap in the learning curve averaged over five rehearsal sessions (i.e., five rehearsal sessions in Fig.1(a) are averaged), so the x-axis reflects 600 iterations.
>
> Thank you again for your valuable comments, questions, and suggestions. We hope our responses have addressed your concerns. Please let us know if you have further questions.

---

### Review · Reviewer_tFwu · 2024-06-09

**Summary Of Contributions:**

The paper addresses the issue of model decay in large pre-trained deep neural networks (DNNs) by employing continual learning (CL). Specifically, the authors identify the stability gap that leads to forgetting in past tasks. They propose two hypotheses that explain the causes of the stability gap and subsequently introduce methods to mitigate this gap.

**Audience:**

Yes

**Claims And Evidence:**

Yes

**Requested Changes:**

+ Discuss the differences between stability gap and forgetting in the literature with more details.

+ Clearly link Sec 4 to Sec 3 and show how the metrics in Sec 3 are used.

**Strengths And Weaknesses:**

Strengths:

+ The paper introduces three insightful metrics: stability gap, plasticity gap, and continual knowledge gap.

+ The experimental results are comprehensive and robustly support the proposed hypotheses.



Weaknesses:

- The motivation of the paper is unclear. The description makes it difficult to distinguish between the stability gap and catastrophic forgetting, as both result in performance drops when introducing new tasks. For instance, it is not evident whether the method proposed in [1] would be applicable in reducing the stability gap.

- The connection between the methods discussed in Section 4 and the theoretical metrics introduced in Section 3 is weak. Although the continual knowledge gap is formally defined with a mathematical formulation, it is not utilized in any subsequent analysis in Section 4.

[1] Kirkpatrick, James, et al. "Overcoming catastrophic forgetting in neural networks." Proceedings of the national academy of sciences 114.13 (2017): 3521-3526.

---

> ### Author Response · Authors · 2024-07-28
> **Author Response**
>
> Thank you for your reviews and insightful feedback. We have edited our paper to address your concerns. Below, we have provided detailed responses to each review separately.
>
> **W1. Is there a difference between forgetting and the stability gap?**
>
> - As discussed in Sec. 2, while forgetting and the stability gap are related in that both involve performance drops in old tasks, they differ in their evaluation. The traditional measures of __catastrophic forgetting__ focus on performance drops __at task transitions__, often overlooking significant forgetting that occurs during the learning process, especially in the early steps [1]. The stability gap captures this.
>
> - __The stability gap__ refers to the performance drops observed over learning steps i.e., __between task transitions__ and can be seen as __transient forgetting__. When performance is measured over time, there is a significant drop in performance on prior tasks as soon as training on a new task begins. This performance then partially recovers through continual learning as the current task is learned.
>
> - We discussed this in Sec. 2. Now, we have explained the distinction more concretely in Fig. 1(a) and the introduction (text is highlighted in blue).
>
> - It has been shown in [1] that various rehearsal- and regularization-based continual learning methods including EWC [2] are vulnerable to the stability gap phenomenon.
>
> **References**:
> - [1] De Lange et. al., "Continual evaluation for lifelong learning: Identifying the stability gap", In ICLR 2023
> - [2] Kirkpatrick, James, et al. "Overcoming catastrophic forgetting in neural networks", Proceedings of the national academy of sciences 114.13 (2017): 3521-3526.
>
> **W2. The connection between evaluation metrics and stability gap mitigation approaches**
>
> - In section 3.2, we formalized our evaluation metrics to measure the stability gap ($S_\Delta$), plasticity gap ($P_\Delta$), and continual knowledge gap ($CK_\Delta$). We found that our proposed mitigation methods in section 4 effectively mitigated the stability, plasticity, and continual knowledge gaps when evaluated using metrics from section 3.2.
> - We tried but it is unclear how to best _link_ those sections since section 3.2 describes the tools to measure the efficacy of the proposed mitigation methods in section 4. Mitigation methods are _designed_ to prevent factors contributing to the stability gap such as large loss at the output layer and excessive network plasticity.
> - All three metrics, including the continual knowledge gap $CK_\Delta$, have been used in all evaluations and experiments in the main paper (Sec. 5, Sec. 6) and the Appendix.
> - Would you please provide more insights on this? We appreciate your suggestion for further clarification and will strive to address this concern.
>
> **Requested Changes**
>
> - **Difference between stability gap and forgetting:** We have included the explanations in the paper and highlighted them in blue.
> - **Link sec 3 to sec 4:** We have tried our best to address it. Please let us know if our explanations address your concerns. We appreciate your feedback and the opportunity to provide clarifications.
>
> Thank you again for your insightful comments, questions, and suggestions. We hope our responses have addressed your concerns. Please let us know if you have further questions.

---

### Review · Reviewer_KC3q · 2024-07-27

**Summary Of Contributions:**

This paper studies the "stability gap phenomenon" in continual learning (CL) that occurs at the early stage of the CL phase, where the accuracy of previous data drops drastically when updating the model with the new data. Normally, the CL method, like rehearsal, will retrain the model with both the old and new data, where the performance on previous data will recover to a satisfactory level after a long-time retraining phase. Hence, the authors consider this phenomenon as a major obstacle to efficient CL. They propose two hypotheses as the possible reason for this phenomenon and a method to reduce the stability gap. The proposed method is tested on large-scale experiments to illustrate its effectiveness.

**Audience:**

Yes

**Claims And Evidence:**

Yes

**Requested Changes:**

1. Add a formal mathematical description for the Class Incremental Learning setting (CIL).
2. In the text above eq(5), $P(x_i;\theta_i)$ is defined as the output softmax of sample $x_i$ from class $k$, I guess it should be $P(y_i=k|x_i;\theta_i)$.
3. Why the softmax probabilities for **each class** $u_k$ is a vector in $R^K$? Isn't it a real value and all the classes form the vector?

**Strengths And Weaknesses:**

### Strengths
1. The paper proposes new metrics to measure the stability, plasticity, and continual knowledge gaps.
2. The stability gap can be mitigated with the proposed weight initialization, soft targets, partly updating the neural networks with LoRA, and further limiting the plasticity by freezing output layer weights of previously seen classes.

### Weaknesses
1. The related work discussion is insufficient, and I am somehow not convinced by the motivation of the work. In my opinion, the stability gap is specifically limited to rehearsal methods, which may not be an obvious problem in other previous methods like continual learning methods combining meta-learning in online fashion [1,2,3] or dynamic architecture methods that create new network pathways or submodules for the new classes [4,5].
2. In addition to the proposed possible reasons (e.g., the large number of parameters to be updated) for the stability gap phenomenon, the learning rate itself can cause a large drop when updating the model with new data. However, such reason is not considered.
3. Some notations are not clear (see required changes), some related setting need more detailed introduction for broader audience, and the writing could be further improved.

[1] Matthew Riemer, Ignacio Cases, Robert Ajemian, Miao Liu, Irina Rish, Yuhai Tu, and Gerald Tesauro. Learning to learn without forgetting by maximizing transfer and minimizing interference. arXiv preprint arXiv:1810.11910, 2018.

[2] Massimo Caccia, Pau Rodriguez, Oleksiy Ostapenko, Fabrice Normandin, Min Lin, Lucas Page-Caccia, Issam Hadj Laradji, Irina Rish, Alexandre Lacoste, David Vázquez, et al. Online fast adaptation and knowledge accumulation (osaka): a new approach to continual learning. Advances in Neural Information Processing Systems, 33:16532–16545, 2020.

[3] Chen, Qi, et al. "On the stability-plasticity dilemma in continual meta-learning: theory and algorithm." _Advances in Neural Information Processing Systems_ 36 (2024).

[4] Yoon, Jaehong, et al. "Lifelong learning with dynamically expandable networks." _arXiv preprint arXiv:1708.01547_ (2017).

[5] Zhenyi Wang, Li Shen, Tiehang Duan, Donglin Zhan, Le Fang, and Mingchen Gao. Learning to learn and remember super long multi-domain task sequence. In Proceedings of the IEEE/CVF Conference on Computer Vision and Pattern Recognition, pages 7982–7992, 2022.

---

> ### Author Response · Authors · 2024-07-28
> **Author Response**
>
> Thank you for reviewing our paper and providing insightful comments and feedback. We have tried to address your concerns by providing explanations below and revising our paper.
>
> **W1. Is the stability gap universal? Could the authors clarify if only rehearsal-based methods exhibit the stability gap?**
>
> - The paper [1] that introduced the stability gap also discussed this topic in supplemental section A. They noted that the stability gap cannot be present in methods that do not permit changes in previously learned tasks. For example, parameter isolation or dynamic architecture methods grow new sub-networks for new tasks while freezing parameters for previous tasks, or they dedicate a model copy to each task [2]. Typically, these methods activate task-specific sub-networks using task labels or learned task classifiers during inference. Consequently, they avoid performance degradation in previously learned tasks but do not allow __backward transfer__. Hence, the stability gap does not apply to these methods. We have explained this in the related work section (Sec. 2). The text is highlighted in blue.
> - Following [1], we studied the stability gap in methods that allow changes in the performance of old tasks, such as rehearsal, regularization, and knowledge distillation methods.
> - In our large-scale continual learning (CL) experiments, we evaluated a variety of CL algorithms: __1) rehearsal-based__ (Vanilla, GDumb), __2) rehearsal and knowledge distillation__ (DERpp), __3) rehearsal-free and knowledge distillation__ (LwF), and __4) online CL__ (REMIND). Our method, SGM, effectively mitigated the stability gap and improved computational efficiency across most common CL methods.
>
> **References:**
>
> - [1] De Lange et. al., "Continual evaluation for lifelong learning: Identifying the stability gap", In ICLR 2023
> - [2] De Lange et. al., "A continual learning survey: Defying forgetting in classification tasks", In TPAMI 2021
>
> **W2. Learning rate was not considered as a factor to cause the stability gap**
>
> - In general, reducing the learning rate favors preserving the old task's performance while increasing it improves new task adaptation.
> We set the learning rate to balance both sides and achieve the best performance.
> - While the learning rate can influence stability or adaptation, it cannot maximize both without other mitigation approaches.
>
> **W3. Could authors further improve notations and settings?**
>
> - We appreciate your feedback. We have made the requested changes. Modified text is highlighted in blue.
> Please let us know if the changes address your concerns.
>
> **Requested Changes**
>
> - **CIL definition:** We have now included a formal mathematical description for the Class Incremental Learning setting (CIL) in Section 3.1.
> - **Equation 5:** We have modified equation 5 and relevant text.
> - **Softmax probabilities:** For __each class__ $k$, we update the entire softmax probabilities over all classes $K$. Therefore, $\mathbf{u}_k \in \mathbb{R}^K$ is a vector. We have included a figure (Fig. 9) illustrating the process in Appendix G. Please let us know if it is still ambiguous. We appreciate your feedback and the opportunity to provide clarifications.
>
> Thank you again for your valuable comments, questions, and suggestions. We hope our responses have addressed your concerns. Please let us know if you have further questions.

---

### Comment · Action_Editor_mVGx · 2024-07-27
**Discussion phase has begun**

Dear authors,

Now the discussion phase begins. Please respond to the reviewers' comments in two weeks.

AE

---

### Author Response · Authors · 2024-09-14
**Camera Ready Submission**

Dear Reviewers and Editors,

Thank you so much for the insightful discussion and time spent reviewing our work. We have made revisions based on the reviewers' suggestions and uploaded a camera ready version. As suggested by the action editor, we have also discussed the connection between our evaluation metrics and stability gap mitigation methods in Appendix H.

Thank you again for your valuable comments and suggestions. We are grateful for the opportunity to publish our work at TMLR.

Kind regards, the authors.

---

### Decision · Action_Editor_mVGx · 2024-09-05

**Recommendation:** Accept as is

**Comment:**

The paper studies an important topic of continual learning and its impact on the model delay occurring in deep learning. The paper identified a new performance metric of the stability gap, based on which the paper proposed novel method to mitigate such a gap and improve the training performance. I think this paper will be a good contribution to the community and hence recommend the acceptance of the paper. One reviewer has a concern on the connection between evaluation metrics and stability gap mitigation approaches. The authors have addressed this in their response. I suggest the authors to add their response to their final version of the paper.

**Audience:**

The topic of continual learning studied in this paper is important for the ML community and will attract substantial interests from TMLR's audience.

**Claims And Evidence:**

Yes. The claims and results made in the submission are well supported by the experiments.